# Historical deforestation drives strong rainfall decline across the southern Amazon basin

Jiangpeng Cui [1] ✉, Shilong Piao [2] ✉, Chris Huntingford [3], Tao Wang [1] & Dominick V. Spracklen [4]

The Amazon forest has recently experienced substantial human-induced loss of forest cover. However, the extent to which such historical deforestation has altered regional observed precipitation through inter-regional atmospheric moisture transport remains unclear. Here, we combine satellite observations and an atmospheric moisture tracking model to quantify these feedbacks over the past four decades (1980-2019). We identify a contrasting northern increase and southern decrease dipole trend in observed precipitation across the Amazon basin. The pronounced reduction in precipitation for the southern Amazon basin reaches up to 3.9-5.4 mm yr$^{-1}$ per year, resulting in an 8-11% decline in annual precipitation across the observation period. We discover that this reduction in precipitation is primarily (52-72%) related to widespread deforestation in the southern basin and upwind regions over South America. Deforestation substantially suppresses forest-sourced moisture, increases atmospheric stability and moisture outflow, leading to precipitation reduction. We also find that climate models substantially underestimate the sensitivity of precipitation to deforestation, implying that the Amazon forest is at risk of major loss much sooner than previously projected.

The Amazon forest is Earth's most biodiverse terrestrial ecosystem (e.g., ref. 1) and is essential in regulating much of the global climate system[2,3]. However, an increasing number of studies suggest that the Amazon forest is approaching a critical threshold beyond which much of it could be irreversibly lost, potentially due to climate change, but may also be initiated by substantial deforestation[4]. Multiple satellite observations show that the Amazon forest has experienced extensive loss of forest cover, particularly in the southern part of the Amazon basin[5,6]. Since the year 1985, natural forest cover has declined by 16%, mainly due to direct human-induced deforestation[7]. The Amazon forest plays a vital role in sustaining regional precipitation by recycling substantial amounts of forest-sourced moisture[3,8–11]. Hence, a deeper understanding on how historical deforestation has altered vegetation-climate moisture feedbacks and related availability of such precipitation recycling is of great importance. Refined knowledge will then

underpin more accurate projections of the future trajectory of the remaining Amazon forest in response to any further deforestation.

Observation-based approaches have already verified that deforestation considerably affects precipitation at small scales in the Amazon basin[12,13]. However, an increasing number of studies suggest that changes in inter-regional atmospheric moisture transport, attributed to large-scale deforestation, likely play a critical role in redistributing forest-sourced moisture and reshaping regional precipitation patterns[8,14–16]. It is relatively straightforward for sophisticated fully coupled land-atmosphere models to simulate how Amazon deforestation alters land-surface evapotranspiration and subsequently moisture transport through atmospheric circulation[14,17,18]. Such models allow factorial simulations to isolate individual effects, but the question still remains whether they are accurately simulated. Extracting the parameterisation of individual processes from data can be more

[1]State Key Laboratory of Tibetan Plateau Earth System, Environment and Resources (TPESER), Institute of Tibetan Plateau Research, Chinese Academy of Sciences, Beijing, China. [2]Institute of Carbon Neutrality, Sino-French Institute for Earth System Science, College of Urban and Environmental Sciences, Peking University, Beijing, China. [3]U.K. Centre for Ecology and Hydrology, Wallingford, Oxfordshire, UK. [4]School of Earth and Environment, University of Leeds, Leeds, UK. ✉e-mail: cuijp@itpcas.ac.cn; slpiao@pku.edu.cn

challenging, as these must be derived from the full-complexity actual system. To quantify the effects of Amazon moisture recycling, algorithms must account for the complex spatial connections between forest-derived moisture sources and precipitation sinks across the region. Fortunately, recent advances in atmospheric moisture-tracking techniques make it possible to trace the trajectories or transport pathways of atmospheric moisture. The use of these algorithms, alongside known changes in rainfall patterns, supports discovering changes in inter-regional moisture transport, which may result from major land use changes[3,15,19,20]. Since the substantial and quantified levels of Amazon basin deforestation in recent decades coincide with a period of available rainfall observations, this presents an opportunity to use such atmospheric moisture tracking to more accurately constrain estimates of how forest cover loss is altering the strength of regional vegetation-climate feedbacks.

The objectives of our study are to explain features of precipitation changes over the Amazon basin and to investigate whether some of the observed changes are linked to direct forest cover change, i.e. deforestation. We first calculate precipitation trends at all locations across the entire Amazon basin for the past four decades (1980–2019) using two observation-based precipitation datasets. We then employ an atmospheric moisture-tracking model[21] that allows us to disentangle the evolving contributions of oceanic versus terrestrial-sourced moisture changes, which together account for the overall observed precipitation trends. Such knowledge of changing driving water fluxes, combined with diagnostics from our atmospheric transport model, can be compared with trends in land cover data[5]. This comparison enables a more rigorous assessment of how both local and upwind forest cover loss, including any

over a large geographical range outside the Amazon basin over South America, impacts local precipitation. Our approach enables the creation of a metric, weighted forest cover, which quantifies these effects ("Methods"). The aim of introducing this metric is to capture all the impacts of any deforestation within and outside the Amazon basin across South America, rather than isolating the impact on rainfall from deforestation within the Amazon basin alone. We mainly focus on the Amazon basin because most deforestation to date has been within the basin.

Unlike most previous studies, we use satellite-based estimates of precipitation and, furthermore, develop a water balance-based estimate of historical evapotranspiration as required as an input to the tracking framework. Creating the latter dataset involves refining satellite-based estimates of evapotranspiration, and hence is also strongly informed by multiple sets of measurements. Both datasets more accurately constrain the driving inputs to our atmospheric tracking framework ("Methods"), reducing uncertainties in trends that may be present in other reanalysis precipitation and evapotranspiration datasets[22]. More importantly, the water balance-based evapotranspiration more accurately captures the signals of deforestation ("Methods"), thereby making it possible to track deforestation-induced change in moisture transport and subsequent to attribute precipitation change.

## Results

### Contrasting north-south precipitation trend and related moisture sources

We first investigate the observed geographical patterns of precipitation trends across the entire Amazon basin for the period 1980–2019, based on two key rainfall datasets (Fig. 1a, b). Although there is no

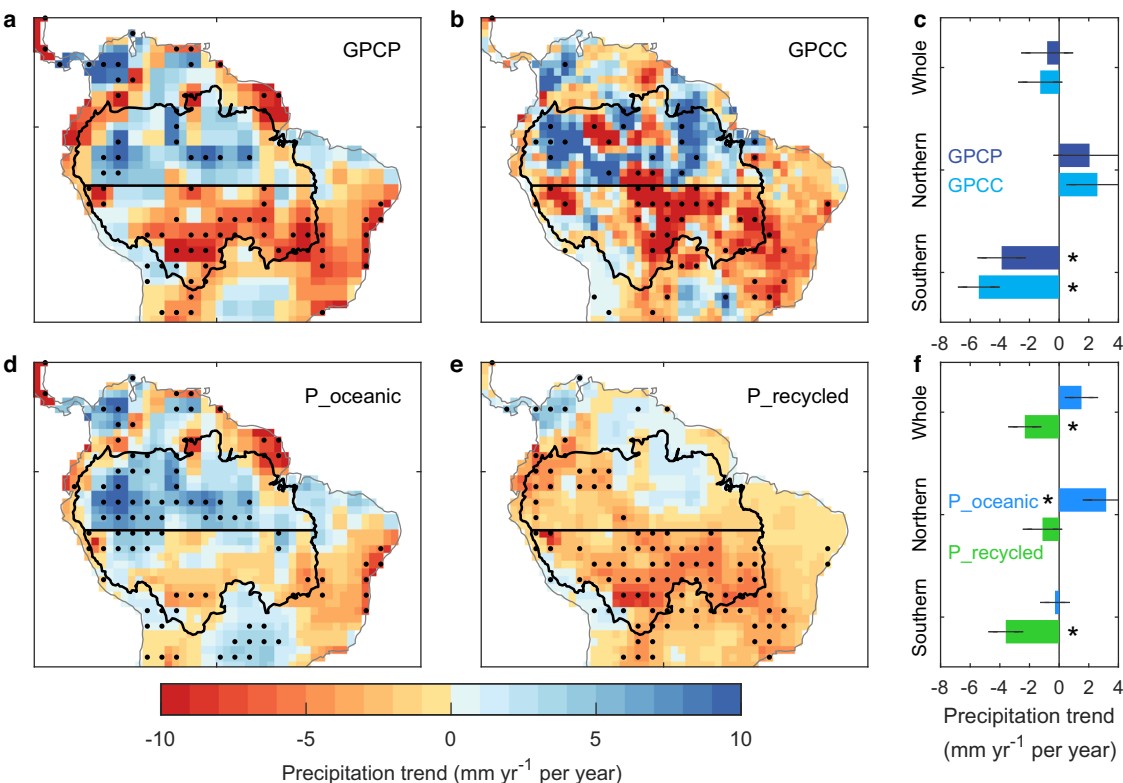

**Fig. 1 | Observed precipitation trend and its moisture sources for the Amazon.** **a** Precipitation trend in the GPCP dataset. **b** Precipitation trend in the GPCC dataset. The horizontal black line, at the latitude of 7.5 °S, indicates our division between the northern and southern Amazon basins, while the outer black curve defines the full spatial extent of the Amazon basin. Stippling is for locations where the trend is statistically significant ($p < 0.05$). **c** Precipitation trend averaged over the whole, northern and southern Amazon basins for the two precipitation datasets. Error bars represent the standard errors of the trends. Asterisks indicate that the trend is significant ($p < 0.05$). **d** Direct oceanic contributions to precipitation trend (P_oceanic). **e** Terrestrial recycled contributions to precipitation trend (P_recycled). **f** Oceanic and terrestrial recycled contributions to precipitation trends averaged over the whole, northern and southern Amazon basins. In all panels, all trends are calculated for the period 1980–2019 inclusively. In (**d**–**f**), P_oceanic and P_recycled are derived from atmospheric moisture tracking based on the GPCP dataset. GPCC-based moisture tracking results are presented in Supplementary Fig. 2. Here, P_total = P_recycled + P_oceanic. Source data are provided with this paper.

overall precipitation trend over the entire basin (Fig. 1c), both datasets are consistent in showing pronounced yet contrasting precipitation trends between the northern and southern parts of the Amazon basin (Fig. 1a, b; a horizontal black line at latitude 7.5°S delineates the divide between the two areas). In the northern basin, precipitation has generally increased over the past four decades (albeit with some very localised decreases). In contrast, most of the southern basin (77–80%) has experienced a major decrease in precipitation. Some areas have demonstrated decreasing precipitation trends that are substantial enough to be statistically significant ($p < 0.05$) and sometimes exceed a lowering of 10 mm yr$^{-1}$ per year. In general, the two precipitation datasets compare well with each other (Fig. 1a versus Fig. 1b), while the remaining local differences in trends are likely related to their differing spatial resolutions and data sources[12,13,23]. Furthermore, both the magnitude and the contrasting north-south pattern of these two precipitation trends generally agree well with other analyses using gauge-based observations[24]. The north-south divergent precipitation trend is more clearly presented when averaged regionally (Fig. 1c). While areal-mean precipitation shows a non-significant increase in the northern basin, it decreases at a statistically significant ($p < 0.05$) rate of 3.9 or 5.4 mm yr$^{-1}$ per year in the southern basin, depending on the two precipitation datasets (Fig. 1c). Notable is that over the past four decades, this statistic corresponds to a very substantial annual reduction in precipitation of 8–11% in the southern portion of the Amazon basin.

We next disentangle the terrestrial recycled from the oceanic contributions to the observed trends in precipitation (Fig. 1d, f) using our simulation structure of atmospheric moisture tracking forced by satellite-based precipitation estimates and water balance-based evapotranspiration calculations ("Methods"). We find that the trends in oceanic-derived precipitation (Fig. 1d) display a similar contrasting north-south pattern of trends in the Amazon basin as the overall measured precipitation trends (Fig. 1a, b). However, importantly, the observed precipitation trends (Fig. 1a, b) cannot be fully explained by the oceanic precipitation trends alone (Fig. 1d). Specifically, the observed decline in precipitation in the southern basin is substantially underestimated when considering only the oceanic-driven precipitation trend. Instead, we observe that the rainfall trends in the southern basin are much better explained when additionally accounting for contributions from terrestrial recycled precipitation (i.e. land-sourced precipitation) trends (Fig. 1e). Indeed, the contribution to the observed precipitation decline in the southern basin is larger from the trends in recycled land precipitation than the trends in the oceanic contribution (Fig. 1e versus Fig. 1d; also Fig. 1f). This finding is also expressed by the statistic of terrestrial recycled precipitation fraction, which takes a high value but has reduced through the observed period (Supplementary Fig. 1). Critically, when averaged across the southern basin, terrestrial recycled precipitation declines by 3.6 or 4.1 mm yr$^{-1}$ per year (dependent on precipitation dataset used), dominating (76% or 92%) the observed overall negative precipitation trend in the southern basin (Fig. 1f and Supplementary Fig. 2). Overall, our results using atmospheric moisture tracking reveal that the strength of land-climate feedbacks of moisture has substantially weakened in the southern Amazon basin over the past four decades. We now investigate our hypothesis that such reductions are linked to direct human influence on the land surface, and particularly deforestation.

## Drivers of precipitation reductions in the southern Amazon basin

To determine the potential underlying land surface drivers responsible for the weakened land-climate moisture feedbacks, we now investigate the trends in key related variables. We include satellite-based forest cover, which represents deforestation (Fig. 2a). We also present the satellite-supported, water balance-derived evapotranspiration used to force the moisture-tracking model (Fig. 2b) (as validated against site and basin-scale measurements from ten sub-basins of the Amazon;

"Methods"). Additionally, we consider solar radiation changes (i.e. downward surface shortwave radiation; Fig. 2c) as a potential forcing. Figure 2a illustrates the widespread loss of forest cover observed in the southern and eastern Amazon basin. In some hotspots, the rate of loss is especially large, exceeding one percentage point per year. Overall, 82% of the southern basin exhibits a negative trend in forest cover. When averaged regionally, the accumulated forest cover loss in the southern basin amounts to 7.7 percentage points over the past 35 years (1982–2016). This substantial decline in forest cover, along with its spatial pattern, aligns closely with negative trends in water balanced-derived evapotranspiration in the southern basin (Fig. 2a versus Fig. 2b, below the horizontal line). These similar patterns strongly suggest, therefore, an important role of forest cover loss in causing the observed decreasing recycled precipitation trend (Fig. 1e) through lower evapotranspiration (Fig. 2b). However, the pronounced reduction of evapotranspiration in the northern basin cannot be attributed to changes in forest cover, as deforestation is markedly less in those locations (Fig. 2a, north of the horizontal line). Instead, the observed decline in solar radiation (Fig. 2c) may explain the reduction in evapotranspiration within the northern basin. Radiation decline exacerbates any inherent energy limitations in these moist northern basin areas, thereby reducing the available energy for evapotranspiration. Other potential drivers, such as rising atmospheric $CO_2$, may also contribute by suppressing vegetation stomatal conductance and thus transpiration[25,26], and future analyses may allow their quantification. However, it is the strongly linked observed spatial patterns in Fig. 2, and in particular comparing Fig. 2a against both Figs. 1e and 2b, that encourage us to investigate the impacts of deforestation further.

In general, deforestation is expected to reshape the spatial pattern of regional precipitation by modulating the properties of both the land surface and the atmosphere. In locations with less human disturbance, forests interact intensively with the atmosphere, enhancing and sustaining the occurrence of precipitation (schematic, Fig. 3a). While small-scale deforestation (less than tens of kilometers) may actually increase local precipitation[12,14,23,27], if large-scale deforestation (of the order a hundred kilometers or greater) occurs (Fig. 3b), such reduced forest cover directly affects precipitation by suppressing the rates of land-atmosphere energy and water exchanges. Specifically, major deforestation is expected to lower levels of available evapotranspiration that drive precipitation. Large-scale deforestation also acts indirectly, increasing atmospheric stability[28] by drying the atmosphere[29] and lengthening the distance of moisture transport, which results in higher moisture outflow and thereby reduces regional precipitation. Additionally, deforestation can also substantially reduce surface roughness, which increases wind speed[30,31], further extending the distance of moisture transport, and causing some moisture to leave the Amazon basin instead.

Our methods, which combine atmospheric transport model with known forcings, open the opportunity to provide a more rigorous and comprehensive understanding of how major deforestation impacts precipitation recycling. However, to test our hypothesis that deforestation suppresses rainfall, we need to advance how we analyse the relationship between changes in forest cover and terrestrial recycled precipitation. Specifically, we need to additionally account for remote upwind deforestation on precipitation (i.e. deforestation in upwind regions that influences local rainfall levels). Therefore, we employ a metric of weighted forest cover, FC_w, which is a statistic ("Methods") derived from the methodology first proposed by Cui et al.[15]. The FC_w variable integrates the satellite-derived forest cover within the combined local and upwind land region of moisture sources (including areas outside the Amazon basin over South America). This integration is weighted by the proportion of land moisture contribution at each upwind location to terrestrial recycled precipitation that falls at each local grid point ("Methods"). Hence, changes in FC_w value capture the full impact of forest cover change, including both locally and upwind, on local precipitation changes.

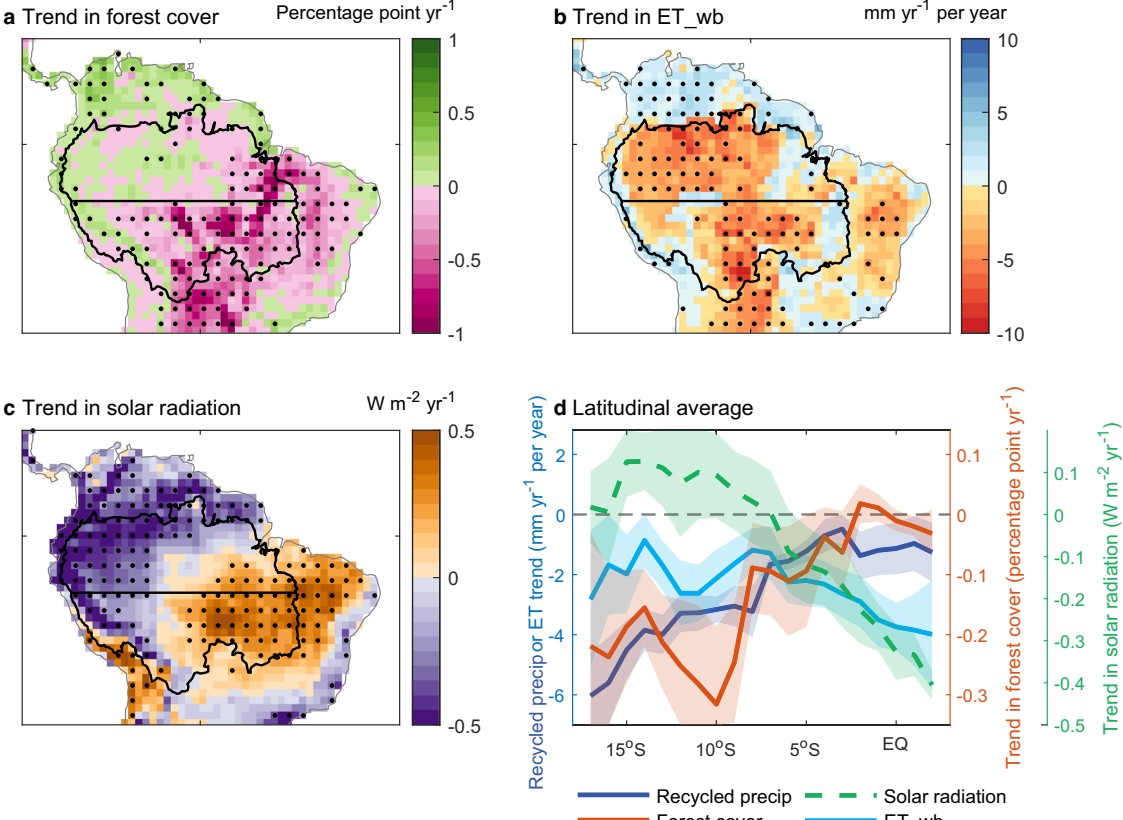

**Fig. 2 | Drivers for the precipitation trend in the southern Amazon basin.**
**a** Trend in forest cover. Stippling indicates locations where the trend is statistically significant ($p < 0.05$). **b** Trend in evapotranspiration, ET_wb, calibrated from water balance calculations. **c** Trend in downward surface solar radiation. **d** Latitudinal averages for the trends in the three drivers presented in (**a**–**c**) and terrestrial recycled precipitation (Fig. 1e). The shaded area indicate the 95% confidence intervals. Source data are provided with this paper.

Our key forest cover dataset spans from the years 1982 to 2016[5]. To align with the years of this coverage, we make these years a common observational period of forest cover and precipitation in the subsequent analysis, noting that precipitation trends show only minor differences during the two periods used (1980–2019 versus 1982–2016; Fig. 1 versus Supplementary Fig. 3). In detail, we utilise measurements and tracking from these 35 years (i.e. 1982–2016) to establish a relationship, incorporating data from locations across the southern Amazon basin, between changes in data-derived terrestrial recycled precipitation and FC_w values (blue dots in Fig. 4a, with the regression line shown in black). The evapotranspiration used to drive the atmospheric moisture tracking are adjusted to eliminate variation across the northern basin, thereby preventing the influence of non-deforestation effects transported from the northern basin on the findings presented in Fig. 4a (see "Methods" for details and rationale). We find a robust correlation between recycled precipitation and changes in FC_w ($R^2 = 0.36$, $p < 0.001$; Fig. 4a). A one percentage point decrease in FC_w reduces local recycled precipitation by 11.6 mm yr$^{-1}$. Overall, the areal mean change in forest cover represented by FC_w, has a decrease of 5.0% during the historical period 1982–2016, corresponding to a reduction in terrestrial recycled precipitation by 96.7 mm yr$^{-1}$ (2.8 mm yr$^{-1}$ per year; solid blue lines in Fig. 4a). This reduction has 95% confidence intervals ranging from −115.6 mm yr$^{-1}$ to −79.0 mm yr$^{-1}$. Such lowering of rainfall (i.e. averaging 2.8 mm yr$^{-1}$ per year) provides our headline statistic that 52–72% of the observed precipitation decline in the southern Amazon basin (3.9–5.4 mm yr$^{-1}$ per year in Fig. 1c) is attributable to deforestation. Therefore, our more sophisticated description of land rainfall recycling, which encapsulates deforestation both locally and upstream via the statistic FC_w, further

supports the conclusion that direct land use is directly contributing to the rainfall reductions shown in Fig. 1e.

To validate the causality of the link between recycled precipitation and FC_w presented in Fig. 4a, we conduct additional process-based experiments using our moisture-tracking model. We first estimate deforestation-induced evapotranspiration changes based on a forest cover-evapotranspiration scaling approach[20]. We then use these estimates, alongside estimates of evapotranspiration with or without deforestation, to directly drive the moisture-tracking model. The difference between these two simulations represents the causal impact of deforestation on recycled precipitation, now calculated directly by evapotranspiration, which represents moisture recycling. The results also show a strong declining (i.e., negative) relationship between recycled precipitation and FC_w (Fig. 4b), supporting our more observational-based results (Fig. 4a). The weaker impact of deforestation on recycled precipitation in Fig. 4b, illustrated by the lower gradient of the fitted regression line of Fig. 4b, may be related to the inclusion of only the direct impact of deforestation on evapotranspiration, thus ignoring feedbacks where atmospheric processes further suppress evapotranspiration and hence precipitation. The differences may also stem from an underestimate of deforestation impacts on evapotranspiration. Understanding these differences in sensitivity may guide future insights or measurement campaigns to better constrain evapotranspiration changes following deforestation.

We extend our analyses further to examine changes in atmospheric dynamics and whether they are potentially linked to alterations in forest cover. We find evidence of decreased Convective Available Potential Energy (CAPE), an increased distance of evapotranspiration moisture transport away from the source, and a reduced

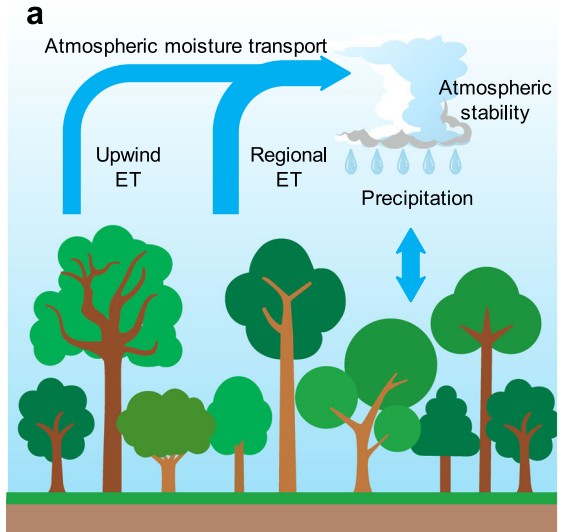

**Fig. 3 | Schematic representation of land-surface and atmospheric processes responsible for the weakening of land-climate feedbacks, due to large-scale deforestation in the southern Amazon basin. a** Intact forest, or regions less disturbed by human activities. These locations feature high and complex canopies which sustains intensive land-atmosphere turbulent mixing and humid air. In these circumstances, regional and upwind evapotranspiration possess strong moisture recycling and feedback mechanisms to maintain regional precipitation. **b** Large-scale deforestation has occurred. In these circumstances, evapotranspiration substantially declines and thus reduces available moisture that feeds into precipitation. Additionally, the drying atmosphere increases its stability, which further reduces

precipitation, lengthens the distance of moisture transport, and promotes moisture flow out of the southern basin (Fig. 5). "Output" represents the atmospheric moisture transported out of a specific region, which here is generally regarded as the Amazon basin. The width of the arrows denotes the relative magnitude of moisture amount in atmospheric transport or land-atmosphere flux exchange. The length of the horizontal part of the arrows represents the relative distance of atmospheric moisture transport. For each process, the corresponding red symbols '+' and '−' in brackets represent an increase or decrease, respectively, in response to deforestation (i.e. the effects in (**b**) compared to those in (**a**)).

evapotranspiration fraction remaining within the local numerical grids in the southern basin (Fig. 5a, c, e; "Methods"). Compared to their climatological mean values (Supplementary Fig. 4), these three variables changed by −21%, 4%, and −19%, respectively, during our study period of 1982–2016, and when averaged over the southern basin. Importantly, all these changes in atmospheric processes are significantly correlated with changes in metrics of forest cover (Fig. 5b, d, f). Additional indirect evidence of raised moisture outflow is found to the south of the Amazon basin (i.e. at latitudes further south than the southern basin), which is characterized by high forest cover loss (Fig. 2a), yet the decline in recycled precipitation (Fig. 1e) remains moderate ($p > 0.05$). This finding indicates an increase in extra moisture transport from the southern Amazon basin to these locations on the edge of the Amazon, offsetting precipitation reduction that might be expected in these areas due to substantial local deforestation. These analyses all support our hypothesis that deforestation substantially reduces regional precipitation by lowering evapotranspiration and increasing atmospheric stability and moisture outflow, as illustrated schematically in Fig. 3.

Our overarching statistic shows that a one percentage point decline in southern basin forest cover (for comparability with other studies only focusing on local deforestation, deforestation outside the southern basin is also added to the local value) results in a 6.0 mm yr⁻¹ reduction in observed annual precipitation, which equates to a fractional 0.32% reduction in the same quantity. However, this observation-based value reveals that climate models generally underestimate the magnitude of the precipitation response to forest cover loss, which is reported to instead have a mean value of 0.16% reduction, according to a meta-analysis study[17]. This underestimation may arise from the inaccurate representation within climate models of the ratio of plant transpiration to total terrestrial evapotranspiration[32,33], affecting calculations where simulated forests are replaced due to deforestation. The underestimation by climate models might also be due to errors in the nonlinear relationship between vegetation-

sourced atmospheric moisture and precipitation variations[34], and atmospheric processes[14,35] which likely understate the sensitivities of land-surface and atmospheric changes to forest cover that we find represented in our Fig. 3. A further study, which is observationally-based, analyzes the local impacts of forest loss[13], by comparing the precipitation differences between neighbouring grids with very different forest cover change. That approach estimates the precipitation response to forest cover loss as 0.25% per percentage point, which is 22% lower than our estimate of 0.32% that includes the full impacts of forest cover loss in both local and upwind regions. This difference reaffirms the importance of accounting for both local and upwind deforestation when assessing impacts on precipitation, and as enabled via our bulk variable FC_w.

### Impacts of future deforestation and mitigation on precipitation

Across the globe, human-driven land cover change is likely to continue into the future, including in the Amazon basin[36]. Hence, we investigate the impact of projected additional forest cover change on precipitation using our established observationally-based linear relationship between changes in recycled precipitation and the FC_w statistic (Fig. 4a; "Methods"). This serves as a reasonable approximation because in the main scenario we consider, SSP2-4.5, climate models project that atmospheric circulation remains roughly unchanged compared to the historical period (see "Methods" and Supplementary Fig. 5). We find that forest cover loss by the end of the 21st century could lead to reductions in annual precipitation of up to 202.4 mm yr⁻¹ in a relatively high-deforestation scenario of primary forest and with no regrowth and mitigation strategies (the SSP2-4.5 "primf" scenario; red lines in Fig. 4a). This rate of deforestation would cause a substantial reduction of about 10.6% of current annual precipitation in the southern Amazon basin (Fig. 4a). A business-as-usual deforestation scenario[37], with faster rates of deforestation that remain similar to those observed in recent decades, would result in a reduction in annual precipitation of up to 15% (−288.1 mm yr⁻¹; Supplementary Fig. 6). It is

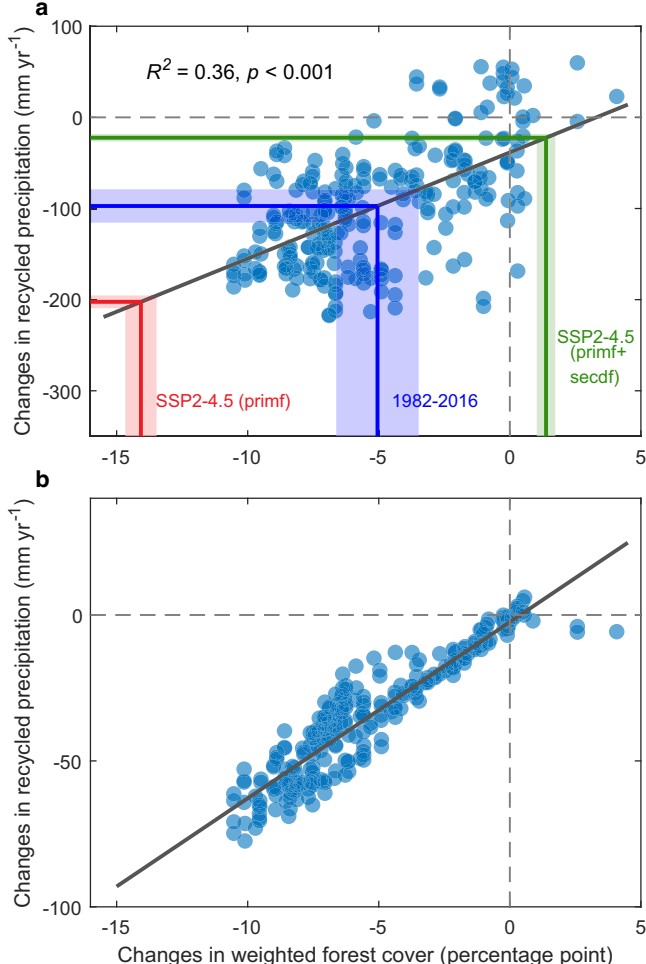

**Fig. 4 | The impacts of forest cover change on recycled precipitation.**
**a** Correlation between weighted forest cover and terrestrial recycled precipitation in the southern Amazon basin. Recycled precipitation is derived from the moisture-trakcing model driven by water balance-based evapotranspiration. Regression line (black line) is based on different spatial points, with each point representing local changes in recycled precipitation and weighted forest cover in the southern Amazon basin for the common period 1982–2016. Each point represents a 1° × 1° gridbox within the southern basin. The blue, red and green lines mark the changes in weighted forest cover in the past 35 years, SSP2-4.5 (primf) and SSP2-4.5 (primf + secdf) scenarios, respectively, and the corresponding reductions in terrestrial recycled precipitation. "Primf" represents primary forested land, while "secdf" represents secondary forested land including forest regrowth and climate mitigation strategies such as afforestation and reforestation ("Methods"). The shaded areas denote the 95% confidence intervals of changes in the southern basin. For illustration purposes, the horizontal and vertical zero lines are shown as grey dashed lines. **b** The same as (**a**), but instead, the level of changes in precipitation caused by altered land moisture recycling is derived from the difference between projections of the moisture-tracking model when driven directly by evapotranspiration estimates with and without deforestation. The evapotranspiration post-deforestation was based on a forest cover-evapotranspiration scaling approach. As (**b**) is direct process model output, we do not present this as a statistical finding (e.g. with $p$ value), but we do fit a linear regression line (black line) to aid comparison with (**a**). Source data are provided with this paper.

particularly noteworthy that these reductions match or even exceed the expected changes to rainfall caused by direct climate change over the same period. Such climate change is due to projected rises in atmospheric greenhouse gases associated with each scenario[38]. As such, a future scenario that instead deliberately includes forest regrowth, and thereby the implementation of climate mitigation strategies such as afforestation and reforestation (SSP2-4.5 primf

+secdf; green lines in Fig. 4a), leads to a reduction in rainfall of only −22.3 mm yr⁻¹. Critically, therefore, forest conservation and afforestation have a major potential to slow down or even reverse any future precipitation reductions caused by higher atmospheric greenhouse gases. Reforestation will strengthen the resilience of the remaining Amazon forest against large-scale dieback risks caused by rainfall reduction due to climate change.

## Discussion

We find a robust correlation between forest cover observations and predictions of rainfall changes using an atmospheric moisture-tracking technique. This suggests that deforestation over the widespread South America, the majority of which has occurred so far in the Amazon basin, substantially reduces observed precipitation across the southern Amazon basin. This reduction in rainfall is caused by decreases in evapotranspiration, which contributes to rainfall and is connected to land use changes. It is also influenced by deforestation-related alterations to inter-regional moisture transport and atmospheric stability, both of which diminish the initiation of rainfall. As our data-driven analysis, using multiple measurement strands, attributes the pronounced recent decline in observed precipitation to large-scale forest cover loss, we therefore strongly corroborate previous modelling studies on deforestation-induced Amazon forest dieback[39,40]. A particular feature of our analysis is the inclusion of the impact of upwind deforestation levels on rainfall feedbacks, via our bulk parameter FC_w. However, we find that climate models, which routinely simulate direct land use changes, tend to underestimate by up to 50% the impact of reduced precipitation caused by large-scale forest cover loss. This finding indicates that current climate model projections of hydroclimatic impacts from deforestation are considerably underestimated in the Amazon basin. Such a lower sensitivity suggests that previous estimates of Amazon tipping points for major forest "dieback" could be reached much sooner than expected, as climate models underestimate the decrease in precipitation caused by deforestation. We note that future changes in global warming[40,41], wildfires[42], drought[43,44] and rising atmospheric $CO_2$ concentrations[25,45], could all have further harmful impacts on the Amazon forest[46]. These may interact strongly with further changes in land use, either directly or through the process of rainfall recycling that we have identified. However, despite these other potential factors, our findings imply that a detailed monitoring of deforestation rates, along with the translation into summary metrics such as FC_w, might be a key component of early warning systems that signal whether the Amazon forest is approaching a tipping point. Alternatively, our research demonstrates that slowing deforestation combined with extensive reforestation could offset the risk of major Amazon dieback caused by climate change, or at least raise the threshold of global warming that could trigger irreversible damage to the forest.

Although we reveal that historical deforestation accounts for much of the observed precipitation reduction in the southern Amazon basin, the availability of more robust long-term observations, such as evapotranspiration[47], vegetation greenness indices[47,48], surface roughness[18], aerosol, and fire smoke[23], will all help refine our findings. Such data will allow an even more accurate evaluation of the impacts of land-surface changes on land-atmosphere water and energy exchanges, as well as atmospheric processes. Hence, more intensive in-situ measurements of vegetation, surface water fluxes and the atmosphere in the Amazon basin will support more tightly constrained assessments of deforestation impact on regional precipitation. Furthermore, it is also essential to develop well-validated coupled land-atmosphere models capable of accurately attributing features of precipitation change to alterations in the land surface or atmosphere[17], and better data will support such an endeavour. This approach may be applicable to models of the Amazon basin only, or to full Earth System Models (ESMs), which remain the primary tool for predicting future large-scale

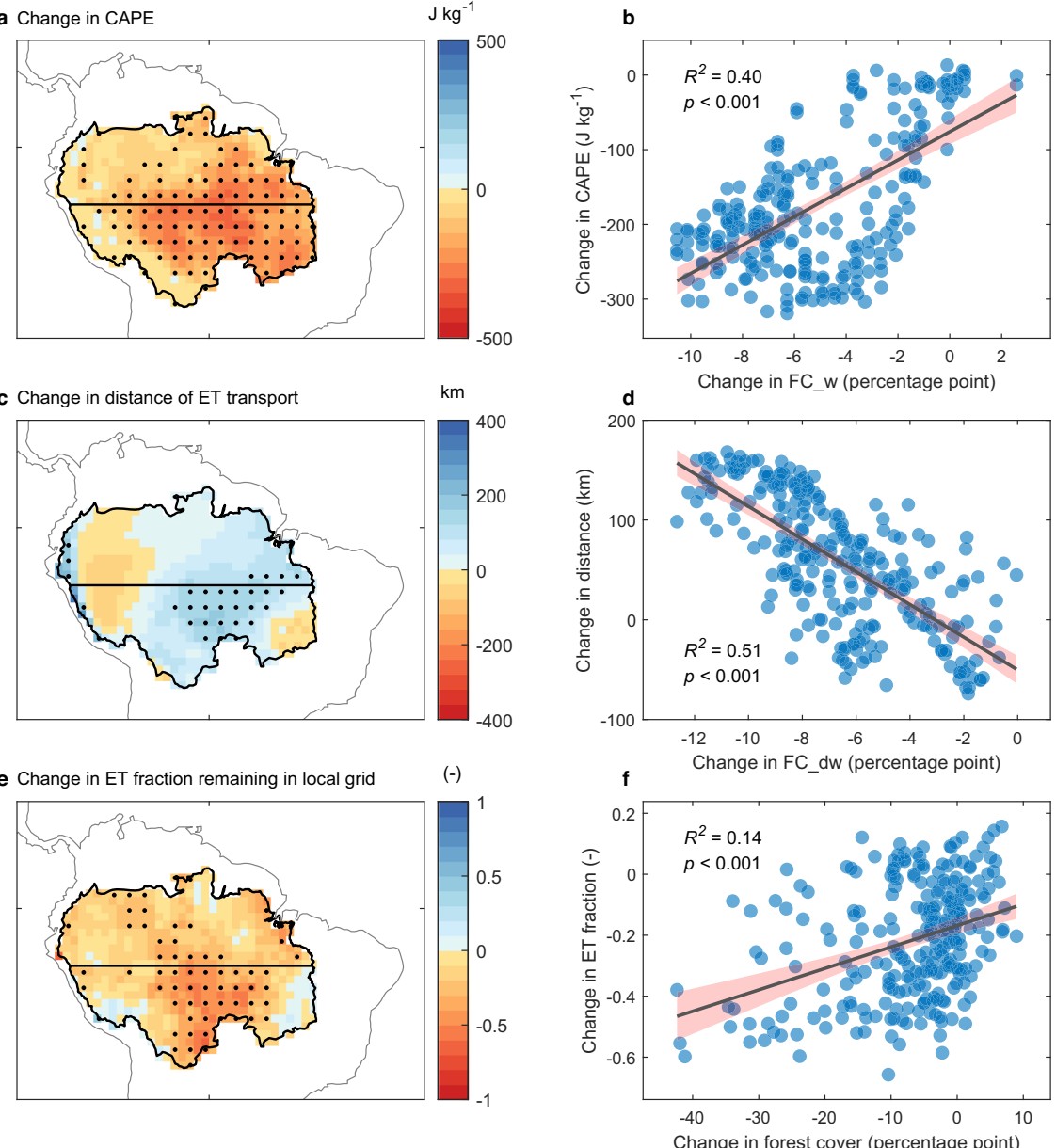

**Fig. 5 | Changes in atmospheric processes and their relationship with forest cover change in the southern Amazon basin. a, c, e** Spatial patterns of **a** changes in Convective Available Potential Energy (CAPE), **c** distance of evapotranspiration (ET) moisture transport and **e** the fraction of moisture from ET remaining in the local grid, all calculated for the period 1982–2016. Stippling indicates regions where the trend is statistically significant ($p < 0.05$). **b, d, f** For individual spatial points, the relationships are shown between changes in **b** CAPE against weighted forest cover (FC_w), **d** distance of ET transport against weighted forest cover in the downwind direction (FC_dw; "Methods"), and **f** the fraction of moisture from ET remaining in the local grid against forest cover. All points in (**b, d, f**) are in the southern Amazon basin, with each point representing a 1° × 1° gridbox, and calculations for the period 1982–2016. In (**b, d, f**), the black lines are a fitted linear regression, and the shaded red areas represent the 95% confidence intervals of the regressions. Source data are provided with this paper.

environmental responses to any future increases in atmospheric greenhouse gases. Although we establish a linear regression, Fig. 4a (with additional process modelling support; Fig. 4b) between precipitation reduction and forest cover loss at a spatial scale based on data from the past 35 years (Fig. 4a), the relationship could become nonlinear at much higher amounts of forest cover loss[12,23,27,34,49]. At much higher deforestation scenarios, the decline in precipitation may be amplified due to either a stronger local self-reinforcing feedback mechanism that accelerates the suppression of recycled rainfall. Additionally, key rainfall thresholds may be crossed, triggering a nonlinear physiological response such that the remaining forest approaches dieback more rapidly. In this context, projected precipitation reductions at the end of the 21st century may be

underestimated, and especially scenarios involving very substantial continued Amazonian deforestation. We note that our quantification of projected precipitation reduction is solely based on changes in terrestrial recycled precipitation and its connection to deforestation over the past decades. Although the changes in the contribution of ocean-sourced moisture to observed precipitation alterations are minor in the historical period and for the southern basin (Fig. 1d, f), there may be more considerable change in the future. In a world much warmer than the present, increasing ocean-sourced moisture could also reshape the patterns of overall precipitation[50,51], which may also affect the fate and timing of the remaining Amazon forest. The projection of future precipitation change can only be considered as a first approximation. Although the relationship between recycled

precipitation and FC_w is statistically significant (Fig. 4a) (further supported by process modelling; Fig. 4b), it does still have sizeable noise, and therefore remaining uncertainty may be amplified with extrapolation. A further caveat is that projected deforestation could induce feedbacks that adjust atmospheric circulations, although this is expected to be relatively small at the larger basin scale[14] ("Methods").

In summary, we find that a highly contrasting north-south trend in observed precipitation has occurred in the Amazon basin over the past four decades. We determine that the pronounced reductions in precipitation across much of the southern basin of the Amazon, including regions still with substantial forest, are primarily driven by large-scale forest cover loss, which overwhelms any other alterations to precipitation caused by climate change. We quantify this land cover feedback in various ways, and with the merging of multiple strands of data, thereby supporting the robustness of findings. A comprehensive and unified overall representation of the process is provided by relating the FC_w metric, which weights and integrates forest cover within the combined local and upwind land regions of moisture sources over South America, to rainfall reductions. In general, previous observational and modelling studies have underestimated reductions in precipitation due to an incorrectly low sensitivity of rainfall to forest cover loss. We find that deforestation substantially weakens the strength of land-climate feedbacks, particularly in the southern basin, mainly by suppressing evapotranspiration but also by increasing atmospheric stability and moisture outflow from the region. We suggest routinely placing the quantification of how forest loss induces rainfall decreases alongside other warnings of Amazon dieback based on the more extensively studied climate change forcings. Climate-induced increases in wildfires and droughts can greatly exacerbate the likelihood of reaching an Amazon forest tipping point, necessitating an understanding of how deforestation feedbacks may further heighten the risk of dieback in remaining forested areas. Conversely, efforts to curb further deforestation and promote forest conservation could enable our identified effect to operate in reverse, serving as a buffer against climate change impacts and thereby reducing the likelihood that the Amazon forest will surpass an irreversible tipping point and dieback.

## Methods
### Observation-based datasets
A summary overview of the various data used in this study is presented in Supplementary Table 1. In more detail, the monthly precipitation data were obtained from the Global Precipitation Climatology Project (GPCP) v.2.3[52]. GPCP precipitation is a merged dataset, incorporating measurements from rain-gauge stations, satellite and sounding observations. The dataset is available at a spatial resolution of $2.5° \times 2.5°$ from 1979 to the present. It is widely used and has been proven to be of high quality[53]. Of relevance to our research, a recent study indicated that the dataset captured well the spatial precipitation pattern induced by deforestation[13]. Additionally, we also used the gauge-based Global Precipitation Climatology Centre (GPCC) precipitation dataset (full data, v2022) as an extra precipitation dataset for comparison. The second dataset has a spatial resolution of $1° \times 1°$, with a temporal period ranging from years 1891 to 2020. Because trends in precipitation averaged over the whole, northern and southern Amazon basin during 1980–2020 were substantially (under)overestimated (−17% ~ +117%) compared to that during periods 1980–2019 or 1980–2018, indicating abnormal high impact of year 2020 (El Niño year) on the long-term precipitation trend (Supplementary Fig. 7), we limited our study period to 1980–2019.

Monthly evapotranspiration was obtained from the Global Land Evaporation Amsterdam Model (GLEAM) v3.5a[54]. This merged model-with-measurements dataset has a spatial resolution of $0.25° \times 0.25°$ and a temporal coverage from 1980 to 2020. The GLEAM system assimilates satellite microwave-based surface-soil moisture

measurements, and vegetation optical depth, and multi-source precipitation observations, which together enable a better constraint on land-surface evapotranspiration[54]. However, existing evapotranspiration products, including the GLEAM, often require the model components to heavily depend on the quality of satellite-derived vegetation indices, which may lead to a poor performance in detecting long-term evapotranspiration trends in the Amazon basin[47,55,56]. To obtain a high-quality evapotranspiration data, we first used a water balance-based method to calculate basin-average evapotranspiration. Then, we obtained ratios between our water balance-based evapotranspiration and basin-average GLEAM evapotranspiration estimates. We then used these calculated ratios to recalibrate GLEAM evapotranspiration predictions across all grids within the basin, so that they match the magnitude of water balance-derived evapotranspiration. In this way, we combined the advantages of the more reliable evapotranspiration trend calculations derived from the water balance method, with the detailed spatial information available from GLEAM evapotranspiration estimates. The water balance-based evapotranspiration was calculated as:

$$ET_{wb} = P - Q - \frac{dS}{dt} \qquad (1)$$

where $P$, $Q$, and $dS/dt$ represent basin-averaged annual precipitation, outlet discharge, and terrestrial water storage ($S$) change, respectively, and all are varying in time, $t$. Annual $dS/dt$ was calculated as the changes in $S$ between two consecutive Decembers. To account for the large-scale spatial heterogeneity of evapotranspiration, we divided the entire Amazon basin into 10 sub-basins according to the locations of hydrological station with long-term discharge observations (Supplementary Fig. 8), and then calculated their water balance-based evapotranspiration for each sub-basin. Monthly discharge data were obtained from the Global Runoff Data Centre (GRDC; Koblenz, Germany). $S$ were obtained from GRACE-REC[57], covering the period from 1979 to 2019 inclusively and with a spatial resolution of $0.5° \times 0.5°$. Although the GRACE-REC does not include the long-term trend of $S$[57], its derived annual change in $S$ ($dS/dt$) has a high correlation ($R^2 = 0.72$) with that from original satellite-based GRACE data[58]. This good comparison suggests that the $dS/dt$ calculation was not substantially impacted by long-term trend of $S$. We also evaluated the performance of our developed evapotranspiration dataset against commonly-used products (i.e., GLEAM[54] and FLUXCOM[59]) and against site-based flux observations[60]. We found that the annual mean $ET_{wb}$ values were consistent with GLEAM and FLUXCOM data at the regional scale. However, the $ET_{wb}$ values showed a more pronounced time-evolving reduction than the latter two datasets (Supplementary Fig. 9). Such larger decreasing trends are consistent with the observed precipitation changes (Fig. 1a, b) and supported by previous observation-based estimates of evapotranspiration[47,58]. At the site level, we also confirmed that our $ET_{wb}$ estimates outperformed the GLEAM and FLUXCOM data products for most locations where point data is available (Supplementary Fig. 10). These advancements in $ET_{wb}$ provided a more solid basis for tracking the dynamics of atmospheric moisture transport, by offering more reliable surface boundary conditions to such models.

In addition to estimates of precipitation and evapotranspiration, we also utilised other satellite-derived products. Forest cover was a fundamental dataset used in our analysis. Long-term forest cover was adopted from the Global Land Change dataset[5]. The dataset was produced by combining optical observations from multiple satellite sensors with a resolution of $0.05° \times 0.05°$ and for data covering a period of 35 years (1982–2016). Trees are defined as all vegetation taller than five meters in height. For our analysis, forest cover is defined as the fraction of a grid covered by the vertical projection of tree crowns[5,6]. Surface solar radiation was obtained from the National Tibetan Plateau Data

Center (TPDC)[61] with a spatial resolution of $10 \times 10$ km (1983–2018). All these observation-based datasets above were resampled to a common spatial resolution of $1° \times 1°$ and at an annual timescale based on the first-order conservative remapping method.

Adding to observation-based datasets, we also obtained surface wind speed estimates from 29 CMIP6 climate models to evaluate the magnitude of wind speed change in the SSP2-4.5 scenario. For this scenario, we compared the mean values for the period from 2081 to 2100 relative to the historical period from 1996 to 2015. The models include ACCESS-CM2, ACCESS-ESM1-5, AWI-CM-1-1-MR, BCC-CSM2-MR, CAMS-CSM1-0, CAS-ESM2-0, CESM2-WACCM, CMCC-CM2-SR5, CMCC-ESM2, CanESM5, EC-Earth3-Veg-LR, EC-Earth3-Veg, EC-Earth3, FGOALS-f3-L, FGOALS-g3, FIO-ESM-2-0, GFDL-ESM4, IITM-ESM, INM-CM4-8, INM-CM5-0, IPSL-CM6A-LR, KACE-1-0-G, MIROC6, MPI-ESM1-2-HR, MPI-ESM1-2-LR, MRI-ESM2-0, NorESM2-LM, NorESM2-MM, and TaiESM1.

## Future forest cover

To explore the impact of future forest cover change on precipitation, we also employed a key scenario of forest cover from the Land-Use Harmonization 2 (LUH2) project, which was designed for the Coupled Model Intercomparison Project Phase 6 (CMIP6)[36]. Forest cover used here includes primary forested land (primf) and secondary forested land (secdf). Quantity primf is defined as natural vegetation that has never been impacted by human activities since the beginning of the LUH2 simulation. Quantity secdf is vegetated land that is recovering from previous human disturbance, and which may include climate mitigation strategies such as afforestation and reforestation. The forest cover change was calculated as the difference between the mean forest cover in the last twenty years of 21st century (2081–2100) and that in a historical baseline (defined as 1996–2015) (Supplementary Fig. 11). Additionally, we also used a regional business-as-usual deforestation scenario, which was generated based on historical deforestation rates and included a realistic deforestation pattern[62] (Supplementary Fig. 6a).

## Atmospheric moisture source and sink tracking

We used a well-established atmospheric moisture tracking model, the Water Accounting Model-2layers (WAM-2layers)[21], to disentangle the oceanic versus terrestrial moisture contributions to the observed precipitation trend across the Amazon basin (see Supplementary Table 2 for an overview of the model). Here, ocean-sourced precipitation (P_oceanic) is defined as the land precipitation that is contributed by moisture from the ocean evaporation, while land-sourced recycled precipitation (P_recycled) is defines as land precipitation that is contributed by moisture from terrestrial evapotranspiration. So observed precipitation (P_total) is the sum of land-sourced and ocean-sourced precipitation (i.e. P_total = P_oceanic + P_recycled). The WAM-2layers is an 2D offline moisture tracking model based on an Eulerian framework, and it quantifies the moisture source-sink relation between precipitation and evapotranspiration by tracking atmospheric moisture forward or backward in time[16]. The major model input includes reanalysis data from the updated ECMWF ERA5 database at a spatial resolution of $1° \times 1°$ for the period 1979–2020. All ERA5-based input variables are 6-h gridded data (vertical specific humidity, zonal and meridional wind speeds, and surface pressure) except for precipitation and evapotranspiration, which have a 1-h temporal resolution. In each 15-min timestep, WAM-2layers solves the water balance of "tagged" moisture in an upper and lower layer in each atmospheric column, and the dynamic and transport of moisture between grids. Because the precipitation inside the Amazon basin can also be affected by forest cover changes outside the basin through cross-regional atmospheric moisture, we tagged moisture from all terrestrial grids to account for moisture changes from both the Amazon basin and outside. This model has proven to perform well against an online fully-3D tracking method[63].

The long-term trends in precipitation and evapotranspiration from ERA5 are known to have uncertainties in the tropics due to sparse gauged-based observations[47,53]. Hence, we used GPCP or GPCC precipitation in the moisture-tracking simulations to replace ERA5 precipitation. Similarly, for evapotranspiration, we employed our water-balance calibrated GLEAM evapotranspiration to substitute ERA5-based evapotranspiration from land, and used OAFlux evapotranspiration to replace ERA5-based evapotranspiration from the ocean[64]. We found that these replacements indeed affected the water balance in WAM-2layers. However, the calculated water loss (imbalance) in our blend of data with these replacements (i.e. ERA5 with observational precipitation and evapotranspiration) was very close to that of the original ERA5 calculations (Supplementary Table 3). These small differences, indicating that the water balance in our study was not overly impacted by the replacement, provide a valuable robustness test. For the use of ERA5 data (precipitation or evapotranspiration), we calculated, for all locations, the ratio between their estimates and observation at the monthly timescale. Then, 3-hourly or 6-hourly ERA data during a month were rescaled proportionally by dividing the ratio[22]. As such, the ERA5 data were scaled by observations at the monthly timescale, while remaining the diurnal cycle, which was necessary to drive the moisture-tracking model. These constraints ensured that the moisture, as tracked from sources to sinks, were consistent with observations over the period of simulations. All these factors together improved the identification of terrestrial recycled precipitation, compared to ocean-sourced precipitation.

To quantify the changes in atmospheric dynamics related to deforestation, we also ran the WAM-2layers model and tracked the eventual sink of evapotranspiration moisture released for each grid individually within the Amazon basin. These calculations were with time-evolving estimates of deforestation ("Methods", above). Through this approach, we calculated the distance of evapotranspiration moisture transport and the evapotranspiration fraction remaining in each grid, along with their changes (Fig. 5). CAPE was obtained from the ERA5 climate reanalysis dataset[65] to represent the atmospheric stability.

For the northern Amazon basin, evapotranspiration decline is not related to forest cover change (Fig. 2a versus Fig. 2b) and may instead be caused by climate change. The inclusion of such non-deforestation related evapotranspiration changes could disturb the regression between changes in forest cover and recycled precipitation. Hence, we performed a factorial simulation with our moisture tracking model to remove the confounding effect of evapotranspiration changes in the northern basin. Over northern basin locations, evapotranspiration was set to be invariant at its seasonal time-average value. This removed the influence of regional climate change that was causing evapotranspiration to decline in the northern basin. These changes were applied to northern basin grids with non-significant LAI decrease (i.e. LAI increases or non-significant LAI decreases). This simulation substantially reduced the uncertainty in the regression in Fig. 4a (R-square: 0.36) compared to the simulation with evapotranspiration change everywhere (0.22 in Supplementary Fig. 12), while did not largely alter the slope of regression (regression slope: 11.6 in Fig. 4a versus 9.3 in Supplementary Fig. 12).

By incorporating atmospheric moisture transport, the development of the process-derived FC_w parameter captured the overall impact of local and upwind deforestation on local precipitation. Therefore, a combination of FC_w and a regression-based model enabled the quantification of how deforestation cross-regionally affects precipitation. We are confident that it is the land surface driving rainfall changes, rather than vice versa, because FC_w accounts for upwind land cover changes and so act as the forcing component. The causality implicit in the weighted parameter FC_w has also been validated in our previous study using a coupled land-atmosphere model[15].

In addition to the WAM-2layers simulations undertaken for our analysis, we obtained another atmospheric moisture tracking dataset

on the fate of land evapotranspiration and precipitation sources developed by Link et al.[66], which was also based on the WAM-2layers[21]. This dataset included the moisture sources for each 1.5° × 1.5° land grid (Precipitationshed)[67] for the period 2001–2018. Precipitationshed (equivalent to "atmospheric watershed") defines the regions where upwind land evapotranspiration or ocean evaporation has contributed to downwind precipitation in the target location[15,67]. The climatological annual precipitationshed (2001–2018 multi-year mean) was used to calculate the weighted forest cover below.

## Weighted forest cover (FC_w)

Precipitation in a specific region is impacted by both local forest cover change and upwind forest cover change, the latter through alterations in moisture transported to the local region via atmospheric transport. Following the methodology proposed by Cui et al.[15], we introduced an aggregated metric of weighted forest cover (FC_w) to account for simultaneously both local and upwind changes. FC_w was calculated for each grid within the Amazon basin by integrating satellite-derived forest cover values within the precipitationshed (here only land surface). This calculation includes weighting by the contribution of each grid in the precipitationshed (sum of the weights equals 1.0; generated by WAM-2layers) to local annual recycled precipitation[15]. Hence, changes to the FC_w statistic contain the full impacts of evolving forest cover change, including both local and upwind cover effects on local precipitation. However, a caveat is that the forest cover dataset used in this study may not capture very fine local-scale (e.g. 30 m) forest cover loss[68] due to its coarse resolution ( ~ 5 km) and extensive cloud cover across the Amazon, obscuring land cover changes[69]. In this sense, deforestation may be underestimated to some extent. In the estimation of the regression between historical weighted forest cover and terrestrial recycled precipitation, the grids located on the border of the northwest South Amazon basin are ignored. This exclusion is because the precipitation there is likely less impacted by forest cover loss in the Amazon basin that instead mainly occurred in the more remote east. In calculating FC_w corresponding to future high-deforestation scenarios and estimated precipitation change, we assumed that future large-scale wind patterns were not substantially altered as previously assumed in earlier studies[8,20,70]. This caveat is reasonable because surface wind speed showed relatively small change (basin average: 2.5%) compared to the historical period at the end of this century in the SSP2-4.5 scenario, as based on 29 different CMIP6 climate models (Supplementary Fig. 5). Furthermore, when considering the impact of forest cover change on wind speed, analysis with a land-atmosphere coupled model indicated that the impact of deforestation-triggered mesoscale atmospheric circulation on precipitation was limited to only 60 km away from the deforested area[14]. This scale is less than the spatial resolution ( ~ 100 × 100 km) of the observation-based datasets and model output used in this study. Bringing these factors together, the FC_w statistic is expected to be reliable to first-order when calculated as the product of future forest cover and the known historical precipitationshed. To estimate the relationship between the distance of evapotranspiration transport and forest cover (Fig. 5d), we also developed another metric of downwind FC_w (FC_dw). Because the moisture transport of evapotranspiration from a specific location was affected by downwind forest cover rather than in the upwind regions, the calculation of FC_dw was the same as FC_w, but instead along the downwind direction.

## Evaluation of autocorrelation impact

FC_w values were derived from forest cover with terrestrial recycled precipitation, weighted by varying amounts depending on the locations within the precipitationshed. Hence, FC_w quantifies the signal of forest cover and its variations, calculated at different locations. However, in our summary Fig. 4a, it is implicitly assumed that the correlation between FC_w and recycled precipitation is not overly affected by the autocorrelation of recycled precipitation. To evaluate the impact of autocorrelation on our results, we introduced a test in three steps (Supplementary Fig. 13). First, we randomly shuffled the grids in the tropics (25°S–15°N) since the forest cover is similar to grids within the Amazon basin, creating a randomized forest cover (rFC). In this way, the spatial information of forest cover pattern was removed. Second, similar to FC_w, we calculated rFC_w based on rFC with recycled precipitation as the weight. As our rFC had no spatial information, the trend in rFC_w was solely driven by the change in recycled precipitation signal. Third, we correlated rFC_w changes with those of recycled precipitation and found the correlation (R-square) to be very low (0.005). Hence, the impact of autocorrelation of recycled precipitation on the correlation in Fig. 4a was limited. This implies that the high correlation in Fig. 4a mainly due to the impact of deforestation on precipitation.

## Data availability

GPCP v2.3 precipitation data are available at https://psl.noaa.gov/data/gridded/data.gpcp.html. GPCC (full data, v2022) precipitation data are available at https://opendata.dwd.de/climate_environment/GPCC/html/download_gate.html. In-situ discharge data are from Global Runoff Data Centre (GRDC; Koblenz, Germany: https://www.bafg.de/GRDC/EN/Home/homepage_node.html). GLEAM) v3.5a evapotranspiration data are available at https://www.gleam.eu/. OAFlux ocean evaporation is available at https://oaflux.whoi.edu. Flux tower observation can be accessed at https://daac.ornl.gov/LBA/guides/CD32_Fluxes_Brazil.html. ERA5 atmospheric and land-surface wind, humidity and fluxes datasets are available at https://www.ecmwf.int/en/forecasts/dataset/ecmwf-reanalysis-v5. Forest cover is available at https://glad.umd.edu/dataset/long-term-global-land-change. TPDC solar radiation is freely access at https://doi.org/10.11888/Meteoro.tpdc.270112. Future land use data are available at https://luh.umd.edu/ and https://daac.ornl.gov/cgi-bin/dsviewer.pl?ds_id=1153. The dataset on the fate of land evapotranspiration and precipitation sources is available at https://doi.org/10.1594/PANGAEA.908705. Projected precipitation recycling is obtained from https://zenodo.org/records/10650579. Wind speed from CMIP6 can be accessed at https://aims2.llnl.gov/search/cmip6/. Source data are provided with this paper.

## Code availability

The codes for WAM-2layers are available via the https://doi.org/10.5281/zenodo.7010594 or at https://github.com/WAM2layers/WAM2layers. The data are processed with Matlab R2021b. The codes for the key methods and Matlab data files related to this work are available at https://doi.org/10.6084/m9.figshare.29649002.v2.

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

## Acknowledgements
This article was supported by the National Natural Science Foundation of China (42522506 and 42471113; J.P.C.) and by the Second Tibetan Plateau Scientific Expedition and Research (STEP) program (2024QZKK0301; J.P.C.). The authors would like to thank Ruud van der Ent for his helpful suggestions on the paper.

## Author contributions
S.L.P. and J.P.C. designed the research; J.P.C. performed the analysis. J.P.C. and C.H. drafted the paper. J.P.C., S.L.P., C.H., T.W. and D.V.S. contributed to the interpretation of the results and to the text.

## Competing interests
The authors declare no competing interests.
