## [Transparent Peer Review file · Nature Communications]

Historical deforestation drives strong rainfall decline across the southern Amazon basin

Corresponding Author: Dr Jiangpeng Cui

Version 0:

Reviewer comments:

Reviewer #1

(Remarks to the Author)

In the revised version of the manuscript, the authors address some of my main concerns, but not others. Unfortunately, I have additional comments that have emerged with the modification of the manuscript. We are getting closer to a version that is publishable, but not yet.

First: The new version of the manuscript has significant grammatical errors. You need to proof-read carefully before resubmitting.

Major Comment 1: Regarding Supplementary Figure R1: thank you for showing the recycling ratio. However, my comment was whether this recycling ratio is changing in time. In other words, when looking at the time series, is the fraction of terrestrial precipitation to total precipitation decreasing in the southern Amazon?

Major Comment 2: Thank you for providing additional details on the bias corrected GLEAM, this is an interesting comparison.

Major Comment 3: Thank you for providing additional analyses on CAPE, moisture transport distance and ET fraction. These are now nicely supporting your conclusions. The increase in distance is particularly interesting. One important effect of deforestation is the decrease in surface roughness which increases wind speed. Studies have shown this is an important mechanism (Lejeune et al. 2015 Climate Dynamics, Eiras-Barca et al. 2020 Annals of the New York Academy of Sciences), but you don't mention this in your paper, please briefly include this potential mechanism.

Major Comment 4: I am now even more confused regarding the experiment described in lines 180-186, shown in Figure 4 and in the methods section.

- You have one simulation with moisture tracking where ET changes everywhere (control)

- You have a second simulation where northern basin ET is maintained at monthly mean value.

This should isolate the effect of the southern basin. How is this linked to Figure 4? How can we see each of the two experiments in this figure? Is each of the blue dots in figure 4 a year and the changing values of Fc correspond to progressive deforestation?

If the goal is to understand how changes in weighted forest cover result in changes in recycled precipitation, then I don't understand why you had to make the northern Amazon ET invariant in the experiment. Why can't you create this figure with only the first experiment?

Major Comment 5: I am satisfied with your response about the CMIP simulations.

Major Comment 6: I like the new title.

Reviewer #2

(Remarks to the Author)

I have a supplementary comment after reading the peer review file:

I totally agree with Reviewer #2, who said, “it seems that the authors were too eager to cast their findings in the mould of Amazon tipping and early warnings, causing their main message to be misleading” To my surprise, this manuscript is the version after a large amount of “Amazon dieback” and “tipping point” content has already been deleted. I think the response to Reviewer #2 is totally insufficient and deserves further revisions, including rediscovering the novelty of the paper, whether for Nature Communications or other journals.

Below are the comments before I read the peer review file:

Cui et al. present an interesting study examining the relationship between rainfall and deforestation across the Amazon Basin using novel methods of water vapor tracking and weighted forest cover. While the topic is of broad interest and the approach is potentially innovative, the manuscript suffers from critical flaws in structure, logic, and clarity, preventing me from recommending it for publication in Nature Communications. Below are my major and minor comments:

Major Comments

1. Disconnection between title, introduction, and results. The manuscript lacks cohesion. The title refers to deforestation’s impact on rainfall, the introduction centers around “Amazon rainforest dieback,” while the results focus on rainfall changes and attribution. These elements are poorly organized and not conceptually aligned.
2. Overemphasis on Amazon dieback: Although Amazon dieback is an important issue, it is not the central topic of this paper. The introduction devotes considerable space to it, with little mention of vegetation–rainfall feedback or deforestation impacts on precipitation. This misalignment continues in the implications section. While dieback may be a consequence, it should not dominate the narrative.
3. Causal logic is flawed: The study investigates rainfall changes and attributes them to various factors, identifying deforestation as a potential driver. However, it incorrectly reverses this conclusion to claim that deforestation drives rainfall changes without robust causal evidence. Moreover, the paper does not even introduce or describe the deforestation dataset used.
4. Misplaced novelty: The true novelty lies in the use of a new water vapor tracking method and the weighted forest cover metric to assess deforestation’s impact on recycled and total rainfall — not in addressing the Amazon dieback issue, which the authors discuss extensively and disproportionately.
5. Misrepresentation of prior literature and conceptual confusion: L12–13: The claim that few studies focus on deforestation’s impact on rainfall is misleading. Many works, e.g., Smith et al. 2023, Qin et al. 2025, have tackled this. L37–39: The phrasing “Amazon forest...through its vegetation” is vague and unscientific. This, along with other imprecise language, suggests a limited understanding of vegetation–climate feedback and a lack of sufficient background review on this key issue.
6. The forest cover dataset is not introduced. Based on my check, it spans 1982–2016, which is inconsistent with the rainfall datasets (1980–2019). This mismatch is not addressed, which undermines the credibility of the analysis. The deforestation should be an important part of this paper, but the authors have even no introductions to it.
7. Weak background and insufficient framing of key concepts. The last paragraph of the introduction fails to clearly state the paper’s objectives. The authors need to better explain: Concepts like “recycled rainfall,” “upwind moisture,” and “weighted forest cover” The novelty and importance of applying water vapor tracking models in studying deforestation–rainfall feedback. Relevant literature on vegetation feedbacks and atmospheric moisture recycling Terms such as “terrestrial recycled precipitation,” “local and regional precipitation,” “upwind deforestation,” and “evapotranspiration” deserve dedicated background discussion. In contrast, the “Amazon dieback” discussion should be limited to one paragraph each in the introduction and the implications.
8. Insufficient methodological detail L566: The description of the WAM-2layer model is overly brief. Key missing details include: What region is “tagged”? All land pixels or only the Amazon Basin? If only Amazon is tagged, how are “recycled” vs. “oceanic” precipitation defined? If all terrestrial pixels are tagged, how are Amazon effects isolated? The distinction between “P_recycled” and “P_oceanic”: Is $P_{total} = P_{recycled} + P_{oceanic}$? How is atmospheric moisture in transit handled? Is the model 2D or 3D in tracking water vapor? These fundamental methodological settings must be clarified.
9. Figure 1 shows that while $P_{recycled}$ is decreasing (linked to deforestation), $P_{oceanic}$ is increasing. However, the paper focuses solely on the decreasing recycled component. This selective focus becomes problematic in Figure 4 and in the discussion, where only $P_{recycled}$ is extrapolated for future projection, which misrepresents the overall hydrological trend and exaggerates the dieback implications.

Minor Comments

1. L486: Clarify why El Niño impacts are discussed for the year 2020.
2. L614–615: The phrase “... LAI increases and non-significant LAI decreases.” is unclear — please revise.
3. L474–487: Why were only two rainfall datasets used? Are they sufficient to support robust conclusions?
4. Figure 4: The use of linear extrapolation should be clearly stated and caveated.
5. Figure 1: “mm yr⁻²” — clarify this unit. It appears inconsistent with “mm yr⁻¹ per year” in the main text. Use consistent and standard units throughout.
6. Figure S4: Label the panels (A, B, C).
7. Figure S6: Label panels A and B.
8. Figure S7: Use consistent labels: panel (a), (b).
9. Figure S12: Label panels A, B, C, D.

Version 1:

Reviewer comments:

Reviewer #1

(Remarks to the Author)

The authors have addressed all of my concerns. The manuscript is now ready for publication.

(Remarks on code availability)

The *.mat files should be included in the public repository so that the results are reproducible.

Reviewer #2

(Remarks to the Author)

The authors have made some textual revisions and provided language-related responses, however these changes lack substantive improvements on several key points. After carefully reviewing the revised manuscript and the authors' responses to each comment, I must flag three critical issues that, in my view, fundamentally undermine both the reliability of the study and the validity of its conclusions:

1. Insufficient and speculative linkage between current results and historical deforestation. The manuscript attempts to link the present results to past deforestation events. However, this argument lacks robust quantitative evidence and relies largely on speculative reasoning. The method used to derive recycled precipitation is disconnected from deforestation, yet the second part of the manuscript makes a subjective determination that the results are induced by deforestation based on "visual correlations". This way does not provide a sound methodological basis for establishing causality. I do not agree with this subjective judgment. As this point is central to the article, I believe that, from this perspective alone, I can no longer support the manuscript.
2. High methodological risk in projecting a weak historical linkage linearly into the future. The manuscript extrapolates a very weak historical relationship between deforestation and precipitation directly into future climate scenarios using a linear approach. This practice poses significant scientific risks and undermines the credibility of the projections.
3. Lack of clear distinction between total terrestrial moisture and Amazonian-sourced moisture. The manuscript does not clearly differentiate between "total terrestrial moisture" and "Amazonian-sourced moisture." Given the stated research aims and study design, the analysis should explicitly tag and track Amazonian-sourced moisture, rather than using the broader "terrestrial moisture" metric. This misuse creates logical inconsistencies and may fundamentally alter the interpretation of the results. For example, by tagging all land grids, the calculated recycled precipitation is artificially elevated, because it includes contributions from land areas outside the Amazon basin. This additional portion has no connection to the Amazonian moisture, deforestation, and future deforestation, yet the authors interpret and present it as "Amazonian-sourced moisture" and use them for attribution explanation and future projection thereby conflating two distinct concepts.

(Remarks on code availability)

Version 2:

Reviewer comments:

Reviewer #2

(Remarks to the Author)

Though the authors make some serious mistakes, I believe this paper is still worthy of publication. However, they need to carefully address the two points raised below. I would also like to see the improvement once the paper is online:

1. In the response to my comment #3, the authors wrote: "The purpose of our study is not to isolate the impact of deforestation solely within the Amazon basin on precipitation. Instead, we aim to identify the role of all deforestation across much of South America, including areas outside the Amazon basin, on historical 'observed' precipitation." If the purpose and results of this study is not specifically focused on the Amazon but rather on South America as a whole, then the title, abstract, and all relevant sections in the manuscript where "Amazon" is used should be revised accordingly.
2. My key concern is that the authors only addressed my comments superficially in the response letter by making language clarifications but made no substantive changes in the main text or results. This way leaves the manuscript with misleading statements and inconsistent conclusions, which undermines the scientific credibility of the work and makes the arguments increasingly self-contradictory. I understand that terms like "Amazon" and the deleted phrase "Amazon dieback" may sound more appealing to editors and readers, but scientific integrity should not be compromised for the sake of attractiveness.

(Remarks on code availability)

Point-to-point responses to reviewer comments

We thank the two reviewers for their particularly careful review and helpful comments/suggestions, all of which have contributed to the development of a better manuscript. Below are the comments from the reviewers, followed by our indented detailed responses in blue. Additional or altered text in the revised manuscript, when cited in this response document, is marked in *dark gray italics*. The line numbers referred to are for the “clean” version of the revised manuscript.

[Reviewer #1 General Comment]

In the revised version of the manuscript, the authors address some of my main concerns, but not others. Unfortunately, I have additional comments that have emerged with the modification of the manuscript. We are getting closer to a version that is publishable, but not yet.

First: The new version of the manuscript has significant grammatical errors. You need to proof-read carefully before resubmitting.

[Response] We are pleased that the reviewer thinks that we are now getting close to a publishable paper. We apologise for not fully addressing all the comments in the first review round, and we are very grateful that you have provided us with a second opportunity to do so here. We appreciate your additional comments. We believe that our revised manuscript has been enhanced owing to your suggestions, and we hope that, based on our responses below, we have now satisfactorily addressed all of the concerns raised.

In this new version, we have also carefully proof-read the whole manuscript to address these grammatical errors and other potential errors.

[Reviewer #1 Major Comment 1]

Major Comment 1: Regarding Supplementary Figure R1: thank you for showing the recycling ratio. However, my comment was whether this recycling ratio is changing in time. In other words, when looking at the time series, is the fraction of terrestrial precipitation to total precipitation decreasing in the southern Amazon?

[Response] Sorry for our misunderstanding. In the revised manuscript, we have redrawn Supplementary Fig. 1 (Fig. R1 below) and added a new panel (b), which shows the time series of the terrestrial precipitation recycling ratio averaged over the southern Amazon basin. As the reviewer expected, we find that the recycling ratio over the southern Amazon basin has significantly decreased in the past four decades. In the main text, we have revised the citation of Supplementary Fig. 1 as: “Indeed, the contribution to the observed precipitation decline in the southern basin is larger from the trends in recycled land precipitation than the trends in the oceanic contribution (Fig. 1e versus Fig. 1d; also Fig. 1f). This finding is also expressed by the statistic of terrestrial recycled precipitation fraction, which takes a high value but has reduced through the observed period (Supplementary Fig. 1)” (Lines 108-112 in the “no track” version).

Fig. R1. (plot appears as Supplementary Fig. 1 in our new version) **Precipitation fraction that contributed from terrestrial recycled moisture.** **a**, Spatial pattern of precipitation fraction that contributed from terrestrial recycled moisture (terrestrial precipitation recycling ratio) over the Amazon basin. The horizontal black line, at the latitude of 7.5°S, indicates the division between the northern and southern Amazon basins, while the outer black curve defines the full spatial extent of the Amazon basin. Gray lines represent the isopleths of major terrestrial recycled precipitation fraction. **b**, Time series of terrestrial precipitation recycling ratio averaged over the southern Amazon basin. The black line is a fitted linear regression, and the shaded red areas represent the 95% confidence intervals of the regressions.

[Reviewer #1 Major Comment 2]

Major Comment 2: Thank you for providing additional details on the bias corrected GLEAM, this is

an interesting comparison.

[Response] Thank you. Your suggestion has helped improve the manuscript, as it has made us better clarify the datasets that feed, either directly or indirectly (as for GLEAM), into the atmospheric transport model.

[Reviewer #1 Major Comment 3]

Major Comment 3: Thank you for providing additional analyses on CAPE, moisture transport distance and ET fraction. These are now nicely supporting your conclusions. The increase in distance is particularly interesting. One important effect of deforestation is the decrease in surface roughness which increases wind speed. Studies have shown this is an important mechanism (Lejeune et al. 2015 Climate Dynamics, Eiras-Barca et al. 2020 Annals of the New York Academy of Sciences), but you don't mention this in your paper, please briefly include this potential mechanism.

[Response] We are pleased that the reviewer finds that presenting these additional diagnostics supports our conclusions more nicely. Also, thank you for providing us with these two very helpful papers. We now cite these papers in the appropriate place, writing: *“Additionally, deforestation can also substantially reduce surface roughness, which increases wind speed^{30,31}, further extending the distance of moisture transport, and causing some moisture to leave the Amazon basin instead”* (Lines 162-164).

[Reviewer #1 Major Comment 4]

Major Comment 4: I am now even more confused regarding the experiment described in lines 180-186, shown in Figure 4 and in the methods section.

- You have one simulation with moisture tracking where ET changes everywhere (control)

- You have a second simulation where northern basin ET is maintained at monthly mean value.

This should isolate the effect of the southern basin. How is this linked to Figure 4? How can we see each of the two experiments in this figure? Is each of the blue dots in figure 4 a year and the changing values of Fc correspond to progressive deforestation?

If the goal is to understand how changes in weighted forest cover result in changes in recycled

precipitation, then I don't understand why you had to make the northern Amazon ET invariant in the experiment. Why can't you create this figure with only the first experiment?

[Response] We are sorry that our introduction to the experiments was not sufficiently clear and indeed, even confusing in places.

Based on this comment, we now include simulations in the paper that do not involve factorial analysis. This diagram represents the “first experiment,” and for reference, it is included in the Supplementary Information as SI Fig. 3 (please see Fig. R2 below).

Removing ET changes in the northern Amazon basin helps avoid the confounding influence of non-deforestation factors, such as climate change, which are considered relevant for this area. In the paper, we show that the ET decline in the northern basin does not align with the relatively minor forest cover changes observed there and is more likely linked to climate change. Therefore, this factorial approach gives us greater confidence that the regression in Fig. 4 is identifying deforestation drivers relevant to the southern basin.

The inclusion of a dynamic ET in the northern basin introduces “noise” to our regression (lower R-square) between changes in forest cover and recycled precipitation (Supplementary Fig. 3 versus Fig. 4), although the regression slopes are similar (regression slope: 11.6 in Fig. 4 versus 9.3 in Supplementary Fig. 3). Therefore, the second factorial experiment, which more effectively isolates deforestation-induced ET from background climate changes, enhances the robustness of the regression in Fig. 4 with a better fit. The consistency of the regression line across the plots supports the idea that Fig. 4. only reflects effects related to the southern basin.

Based on your request and the points raised above, we have revised the method description in the main text as: *“We utilise measurements and tracking from these 35 years (i.e. 1982-2016) to establish a relationship, incorporating data from locations across the southern Amazon basin, between changes in data-derived terrestrial recycled precipitation and FC_w values (blue dots in Fig. 4, with the regression line shown in black). The evapotranspiration used to drive the atmospheric moisture tracking are adjusted to eliminate variation across the northern basin,*

thereby preventing the influence of non-deforestation effects transported from the northern basin on the findings presented in Fig. 4 (see Methods for details and rationale)” (Lines 185-192).

In the Methods section, we now provide a more detailed and comprehensive introduction to our experiments, and in particular, we aim to eliminate any risk of ambiguity. We write text: “For the northern Amazon basin, evapotranspiration decline is not related to forest cover change (Fig. 2a versus Fig. 2b) and may instead be caused by climate change. The inclusion of such non-deforestation related evapotranspiration changes could disturb the regression between changes in forest cover and recycled precipitation. Hence, we performed a factorial simulation with our moisture tracking model to remove the confounding effect of evapotranspiration changes in the northern basin. Over northern basin locations, evapotranspiration was set to be invariant at its seasonal time-average value. This removed the influence of regional climate change that was causing evapotranspiration to decline in the northern basin. These changes were applied to northern basin grids with non-significant LAI decrease (i.e. LAI increases or non-significant LAI decreases). This simulation substantially reduced the uncertainty in the regression in Fig. 4 (R-square: 0.36) compared to the simulation with evapotranspiration change everywhere (0.22 in Supplementary Fig. 12), while did not largely alter the slope of regression (regression slope: 11.6 in Fig. 4 versus 9.3 in Supplementary Fig. 12)” (Lines 635-648).

Sorry for the initially incomplete caption to Fig. 4. Each blue dot in Fig. 4 represents a grid box within the southern basin. We now clearly state this in the caption, as: “Regression line (black line) is based on different spatial points, with each point representing local changes in terrestrial recycled precipitation and weighted forest cover in the southern Amazon basin for the common period 1982-2016. Each point represents a $1^{\circ} \times 1^{\circ}$ gridbox within the southern basin” (Lines 832-835).

Fig. R2. (plot appears as Supplementary Fig. 12 in our new version) **The impacts of forest cover change on recycled precipitation.** Spatial relationship (black line) between changes in terrestrial recycled precipitation and weighted forest cover in the southern Amazon basin for the period 1982-2016. Each point represents a $1^\circ \times 1^\circ$ gridbox within the southern basin. Different from Fig. 4, recycled precipitation from WAM-2layers here is driven by evapotranspiration allowed to change everywhere (including the northern basin).

[Reviewer #1 Major Comment 5]

Major Comment 5: I am satisfied with your response about the CMIP simulations.

[Response] Thank you, and we are grateful for your original suggestion, which enhanced the robustness through improved use of the CMIP simulations.

[Reviewer #1 Major Comment 6]

Major Comment 6: I like the new title.

[Response] Thank you.

[Reviewer #2 Supplementary Comment]

I have a supplementary comment after reading the peer review file:

I totally agree with Reviewer #2, who said, “it seems that the authors were too eager to caste their findings in the mould of Amazon tipping and early warnings, causing their main message to be misleading” To my surprise, this manuscript is the version after a large amount of “Amazon dieback” and “tipping point” content has already been deleted. I think the response to Reviewer #2 is totally insufficient and deserves further revisions, including rediscovering the novelty of the paper, whether for Nature Communications or other journals.

[Response] First, we thank the new Reviewer #2, standing in for our original Reviewer #2. We truly appreciate your time in evaluating our manuscript.

As you point out, the original Reviewer #2 was very keen on removing any mention of tipping points. All authors have reflected on this, and we agree. In particular, we decided that our main finding (Figure 4) is broadly linear. That is, for the level of weighted changes we consider in forest cover, such changes relate linearly to recycled precipitation. Although at even higher levels of deforestation, nonlinearities (or even tipping points) may emerge, we cannot confirm this with current data. Therefore, the version you reviewed has been adjusted with that framing in mind.

Nevertheless, although our findings are within a “linear regime,” our results, supported by extensive data combined with descriptions of atmospheric moisture recycling, show significant rainfall recycling losses due to deforestation. This remains the main novelty of our analysis.

Following this reviewer's comment, we have tightened the sentences in the Abstract and extensively rewritten the Introduction and Discussion sections. We have reduced the content on an eventual potential tipping point associated with a rapid forest dieback to a comment in the Discussion. This leaves the remaining text focused on vegetation-climate feedback whereby deforestation impacts downwind precipitation, which is the novelty of our study. Please see our response to your Major Comment #2 for details of the revised Introduction and Discussion, and Major Comment #4 “Misplaced novelty” for more correctly representing our novelty.

[Reviewer #2 General Comment]

Below are the comments before I read the peer review file:

Cui et al. present an interesting study examining the relationship between rainfall and deforestation across the Amazon Basin using novel methods of water vapor tracking and weighted forest cover. While the topic is of broad interest and the approach is potentially innovative, the manuscript suffers from critical flaws in structure, logic, and clarity, preventing me from recommending it for publication in Nature Communications. Below are my major and minor comments:

[Response] Thank you for your initial comments. We are pleased that the work is described as “novel”, “innovative”, and of “broad interest”. We now recognise that in some parts of the original submitted manuscript, the logic was not explained as clearly as it should be. We trust the concept of “weighted forest cover” as a novel bulk parameter that we verify allows for an accurate characterisation of suppressed rainfall feedbacks from deforestation. However, we accept that a better organisation is needed to present the logic in an unambiguous and transparent way.

Therefore, we have thoroughly reviewed the whole paper, ensuring that each part clearly presents the logic and is complete. We sincerely hope the revised manuscript is found acceptable in this regard.

[Reviewer #2 Major Comment 1]

1. Disconnection between title, introduction, and results. The manuscript lacks cohesion. The title refers to deforestation’s impact on rainfall, the introduction centers around “Amazon rainforest dieback,” while the results focus on rainfall changes and attribution. These elements are poorly organized and not conceptually aligned.

[Response] Thank you for pointing this out. We acknowledge that the original paper version had a poorly written storyline, resulting in a disconnection between the Title, Introduction, and results. We regret this initial poor writing, as the technical components do form a consistent and

linked set of findings. The analysis leads to the overall aim of our study, which is the quantification of precipitation changes and their attribution. Using two independent precipitation datasets, we identify a contrasting dipole trend, with a northern increase and a southern decrease, across the Amazon basin. Further analyses suggest that the decline in precipitation in the southern basin is due to extensive deforestation, as identified through the combination of atmospheric moisture tracking methods and satellite-based forest cover data.

To improve the coherence of our storyline, we have considerably reduced the content on rainforest dieback, which is more speculative should there be further deforestation and related rainfall suppression. Dieback risk is now briefly mentioned in the Discussion, maintaining a focus on the critical role of deforestation in driving precipitation reduction in the southern Amazon basin.

Please see our response to your Major Comment #2 below for full details of the rewritten text.

[Reviewer #2 Major Comment 2]

2. Overemphasis on Amazon dieback: Although Amazon dieback is an important issue, it is not the central topic of this paper. The introduction devotes considerable space to it, with little mention of vegetation–rainfall feedback or deforestation impacts on precipitation. This misalignment continues in the implications section. While dieback may be a consequence, it should not dominate the narrative.

[Response] Thank you. Deforestation-related reductions in rainfall may have numerous implications, beyond any potential risk of triggering “dieback” for the remaining forest. In the revised manuscript, we have rewritten the Introduction section, eliminated nearly all content on rainforest dieback, and highlighted the importance of quantifying vegetation-rainfall feedbacks and the potential for their modulation by deforestation. Although the following is a lengthy excerpt of new text, for completeness, we present the extensively revised parts of the Introduction.

“The Amazon forest is Earth’s most biodiverse terrestrial ecosystem (e.g., ref. ¹) and is essential

in regulating much of the global climate system^{2,3}. However, an increasing number of studies suggest that the Amazon forest is approaching a critical threshold beyond which much of it could be irreversibly lost, potentially due to climate change, but may also be initiated by substantial deforestation⁴. Multiple satellite observations show that the Amazon forest has experienced extensive loss of forest cover, particularly in the southern part of the Amazon basin^{5,6}. Since the year 1985, natural forest cover has declined by 16%, mainly due to direct human-induced deforestation⁷. The Amazon forest plays a vital role in sustaining regional precipitation by recycling substantial amounts of forest-sourced moisture^{3,8-11}. Hence, a deeper understanding on how historical deforestation has altered vegetation-climate moisture feedbacks and related availability of such precipitation recycling is of great importance. Refined knowledge will then underpin more accurate projections of the future trajectory of the remaining Amazon forest in response to any further deforestation.

Observation-based approaches have already verified that deforestation considerably affects precipitation at small scales in the Amazon basin^{12,13}. However, an increasing number of studies suggest that changes in inter-regional atmospheric moisture transport, attributed to large-scale deforestation, likely play a critical role in redistributing forest-sourced moisture and reshaping regional precipitation patterns^{8,14-16}. It is relatively straightforward for sophisticated fully coupled land-atmosphere models to simulate how Amazon deforestation alters land-surface evapotranspiration and subsequently moisture transport through atmospheric circulation^{14,17,18}. Such models allow factorial simulations to isolate individual effects, but the question still remains whether they are accurately simulated. Extracting the parameterisation of individual processes from data can be more challenging, as these must be derived from the full-complexity actual system. To quantify the effects of Amazon moisture recycling, algorithms must account for the complex spatial connections between forest-derived moisture sources and precipitation sinks across the region. Fortunately, recent advances in atmospheric moisture-tracking techniques make it possible to trace the trajectories or transport pathways of atmospheric moisture. The use of these algorithms, alongside known changes in rainfall patterns, supports discovering changes in inter-regional moisture transport, which may result from major land use changes^{3,15,19,20}. Since the substantial and quantified levels of

Amazon basin deforestation in recent decades coincide with a period of available rainfall observations, this presents an opportunity to use such atmospheric moisture tracking to more accurately constrain estimates of how forest cover loss is altering the strength of regional vegetation-climate feedbacks.

The objectives of our study are to explain features of precipitation changes over the Amazon basin and to investigate whether some of the observed changes are linked to direct forest cover change, i.e. deforestation. We first calculate precipitation trends at all locations across the entire Amazon basin for the past four decades (1980-2019) using two observation-based precipitation datasets. We then employ an atmospheric moisture tracking model²¹ that allows us to disentangle the evolving contributions of oceanic versus terrestrial-sourced moisture changes, which together account for the overall observed precipitation trends. Such knowledge of changing driving water fluxes, combined with diagnostics from our atmospheric transport model, can be compared with trends in land cover data⁵. This comparison enables a more rigorous assessment of the impact of local and upwind forest cover loss on local precipitation, allowing the creation of a new metric of weighted forest cover that quantifies these effects (Methods). Unlike most previous studies, we use satellite-based estimates of precipitation and, additionally, develop a new water balance-based estimate of historical evapotranspiration as required as an input to the tracking framework. Creating the latter dataset involves refining satellite-based estimates of evapotranspiration, and hence is also strongly informed by measurements. Both datasets more accurately constrain the driving inputs to our atmospheric tracking framework (Methods), removing uncertainties in trends that may be present in other reanalysis precipitation and evapotranspiration datasets²² (Lines 17-71, in the “no track” version).

We also include the revised text in the Discussion section below. Once again, we emphasise that our focus now is much more on quantifying and directly addressing the consequences of deforestation-induced rainfall decline, rather than including more speculative comments about the potential for rainforest dieback. As the changes are quite extensive, we apologise for the length of the repeated text below, but we wanted to show all changes.

“Discussion

We find a robust correlation between forest cover observations and predictions of rainfall changes using an atmospheric moisture-tracking technique. This suggests that deforestation in the southern Amazon and upwind regions substantially reduces observed precipitation across the southern basin. The moisture-tracking method includes a forcing with evapotranspiration derived from a water-balance model, which is independently derived from the land use observations, ensuring the two data strands in the correlation independent. This reduction in rainfall is caused by decreases in evapotranspiration, which contributes to rainfall and is connected to land use changes. It is also influenced by deforestation-related alterations to inter-regional moisture transport and atmospheric stability, both of which diminish the initiation of rainfall. As our data-driven analysis, using multiple measurement strands, attributes the pronounced recent decline in observed precipitation to large-scale forest cover loss, we therefore strongly corroborate previous modelling studies on deforestation-induced Amazon forest dieback^{39,40}. A particularly novel feature of our analysis is the inclusion of the impact of upwind deforestation levels on rainfall feedbacks, via our bulk parameter FC_w . However, we find that climate models, which routinely simulate direct land use changes, tend to underestimate by up to 58% the impact of reduced precipitation caused by large-scale forest cover loss. This finding indicates that current climate model projections of hydroclimatic impacts from deforestation are considerably underestimated in the Amazon basin. Such a lower sensitivity suggests that previous estimates of Amazon tipping points for major forest “dieback” could be reached much sooner than expected, as climate models underestimate the decrease in precipitation caused by deforestation. We note that future changes in global warming^{40,41}, wildfires⁴², drought^{43,44} and rising atmospheric CO₂ concentrations^{25,45}, could all have further harmful impacts on the Amazon forest⁴⁶. These may interact strongly with further changes in land use, either directly or through the process of rainfall recycling that we have identified. However, despite these other potential factors, our findings imply that a detailed monitoring of deforestation rates, along with the translation into summary metrics such as FC_w , might be a key component of early warning systems that signal whether the Amazon forest is approaching a tipping point. Alternatively, our research demonstrates that slowing deforestation combined with extensive reforestation could offset the risk of major Amazon dieback caused by climate

change, or at least raise the threshold of global warming that could trigger irreversible damage to the forest” (Lines 272-302).

[Reviewer #2 Major Comment 3]

3. Causal logic is flawed: The study investigates rainfall changes and attributes them to various factors, identifying deforestation as a potential driver. However, it incorrectly reverses this conclusion to claim that deforestation drives rainfall changes without robust causal evidence. Moreover, the paper does not even introduce or describe the deforestation dataset used.

[Response] We apologise for this misunderstanding regarding causality, which mainly arises from our failure to clearly outline the methodology initially.

Our study examines rainfall variations downwind of various locations within the Amazon, using an atmospheric transport model. We then find, through independent data sources, that these precipitation differences are closely linked to the levels of deforestation occurring at or upwind of those sites. Our explanatory model depends heavily on the detailed parameterisation of local and upwind rainfall recycling changes driven by deforestation, achieved through the “weighted forest cover parameter, FC_w” (Cui et al., 2022). The calculation of this parameter is derived directly from process understanding, including atmospheric moisture transport effects (Spracklen et al., 2012). Although our Fig. 4 is regression-based, the inclusion of process-derived parameter FC_w in that regression allows us to bring a substantial focus on causal inference.

For much research on rainfall patterns, this is usually associated with understanding how climate-induced changes to precipitation may impact terrestrial ecosystems. Here, we indeed reverse that concept, finding that Amazon deforestation in the southern basin is affecting rainfall patterns. We are confident of this, because the FC_w statistic takes account, explicitly, of rainfall at, and critically, upwind of locations. This finding effectively defines the causal chain as starting with the land surface.

Therefore, we clarified the causal logic of our study further, writing: “*By incorporating*

atmospheric moisture transport, the development of the process-derived FC_w parameter captured the overall impact of local and upwind deforestation on local precipitation. Therefore, a combination of FC_w and a regression-based model enabled the quantification of how deforestation cross-regionally affects precipitation. We are confident that it is the land surface driving rainfall changes, rather than vice versa, because FC_w accounts for upwind land cover changes and so act as the forcing component. The causality implicit in the weighted parameter FC_w has also been validated in our previous study using a coupled land-atmosphere model¹⁵ (Lines 663-669).

We emphasise that the forest cover and recycled precipitation (including its forcings), shown as the “x” and “y” axes in Fig. 4, are independently derived from satellite observations and moisture-tracking models, respectively. The high correlation between the two variables strongly suggests a causal link between deforestation and reductions in precipitation. In the discussion section, we have added: “*We find a robust correlation between forest cover observations and predictions of rainfall changes using an atmospheric moisture-tracking technique. This suggests that deforestation in the southern Amazon and upwind regions substantially reduces observed precipitation across the southern basin. The moisture-tracking method includes a forcing with evapotranspiration derived from a water-balance model, which is independently derived from the land use observations, ensuring the two data strands in the correlation independent*” (Lines 272-277).

Unfortunately, text was accidentally omitted during the development of the initial paper version, and we apologise for this. We now provide a full description of the deforestation dataset in the main text as: “*Our key forest cover dataset spans from the years 1982 to 2016⁵. To align with the years of this coverage, we made these years a common observational period of forest cover and precipitation in the subsequent analysis, noting that precipitation trends show only minor differences during the two periods used (1980-2019 versus 1982-2016; Fig. 1 versus Supplementary Fig. 3)*” (Lines 181-185).

Furthermore, we have also included more detailed information on this forest cover dataset in the Methods section as: “*Forest cover was a fundamental dataset used in our analysis. Long-*

term forest cover was adopted from the Global Land Change dataset⁵. The dataset was produced by combining optical observations from multiple satellite sensors with a resolution of $0.05^\circ \times 0.05^\circ$ and for data covering a period of 35 years (1982-2016). Trees are defined as all vegetation taller than five meters in height. For our analysis, forest cover is defined as the fraction of a grid covered by the vertical projection of tree crowns^{5,6}” (Lines 547-552).

[Reviewer #2 Major Comment 4]

4. Misplaced novelty: The true novelty lies in the use of a new water vapor tracking method and the weighted forest cover metric to assess deforestation's impact on recycled and total rainfall — not in addressing the Amazon dieback issue, which the authors discuss extensively and disproportionately.

[Response] As outlined above, we considered the dieback issue, and we agree that it is a more speculative aspect of our work that detracts from our very definite and identified current links between deforestation and terrestrial rainfall recycling. Therefore, we have removed nearly all references to potential forest dieback in the Introduction and Discussion sections. We now more clearly state the novelty of our study, which is achieved with the new moisture tracking method and the weighted forest cover, in various parts of our study as:

“It is relatively straightforward for sophisticated fully coupled land-atmosphere models to simulate how Amazon deforestation alters land-surface evapotranspiration and subsequently moisture transport through atmospheric circulation^{14,17,18}. Such models allow factorial simulations to isolate individual effects, but the question still remains whether they are accurately simulated. Extracting the parameterisation of individual processes from data can be more challenging, as these must be derived from the full-complexity actual system. To quantify the effects of Amazon moisture recycling, algorithms must account for the complex spatial connections between forest-derived moisture sources and precipitation sinks across the region. Fortunately, recent advances in atmospheric moisture-tracking techniques make it possible to trace the trajectories or transport pathways of atmospheric moisture. The use of these algorithms, alongside known changes in rainfall patterns, supports discovering changes in inter-regional moisture transport, which may result from major land use changes^{3,15,19,20}” (Lines 35-47).

“Such knowledge of changing driving water fluxes, combined with diagnostics from our atmospheric transport model, can be compared with trends in land cover data⁵. This comparison enables a more rigorous assessment of the impact of local and upwind forest cover loss on local precipitation, allowing the creation of a new metric of weighted forest cover that quantifies these effects” (Lines 59-64).

“We find a robust correlation between forest cover observations and predictions of rainfall changes using an atmospheric moisture-tracking technique. This suggests that deforestation in the southern Amazon and upwind regions substantially reduces observed precipitation across the southern basin. The moisture-tracking method includes a forcing with evapotranspiration derived from a water-balance model, which is independently derived from the land use observations, ensuring the two data strands in the correlation independent” (Lines 272-277).

[Reviewer #2 Major Comment 5]

5. Misrepresentation of prior literature and conceptual confusion: L12–13: The claim that few studies focus on deforestation’s impact on rainfall is misleading. Many works, e.g., Smith et al. 2023, Qin et al. 2025, have tackled this. L37–39: The phrasing “Amazon forest...through its vegetation” is vague and unscientific. This, along with other imprecise language, suggests a limited understanding of vegetation–climate feedback and a lack of sufficient background review on this key issue.

[Response] We apologise for any confusion. In this study, we combine observations of forest cover with atmospheric moisture-tracking methods to quantify the impact of deforestation on observed rainfall via inter-regional moisture transport. We clarify our aim by explicitly stating that we address this omission as: *“However, it remains unclear the extent to which such historical deforestation has altered regional observed precipitation through inter-regional atmospheric moisture transport” (Lines 2-4).*

Smith et al. (2023) quantified the impact of deforestation on local precipitation using observational datasets, while Qin et al. (2025) investigated the non-local effects of

deforestation on seasonal precipitation changes based on a coupled land-atmosphere model. These are key papers, and thank you for reminding us of this. We now include these two references in the main text, along with other further relevant papers as: “*Observation-based approaches have already verified that deforestation considerably affects precipitation at small scales in the Amazon basin*^{12,13}. However, an increasing number of studies suggest that changes in inter-regional atmospheric moisture transport, attributed to large-scale deforestation, likely play a critical role in redistributing forest-sourced moisture and reshaping regional precipitation patterns^{8,14-16}” (Lines 31-35).

For L37-39, this sentence has been improved as “*The Amazon forest plays a vital role in sustaining regional precipitation by recycling substantial amounts of forest-sourced moisture*^{3,8-11}” (Lines 24-25 in the “no track” version).

In the revised manuscript, we have conducted a more detailed background literature review and placed an enhanced emphasis on vegetation-climate feedbacks and the impact of deforestation on precipitation. This relates to our response above to your Major Comment #2 for details.

[Reviewer #2 Major Comment 6]

6. The forest cover dataset is not introduced. Based on my check, it spans 1982–2016, which is inconsistent with the rainfall datasets (1980–2019). This mismatch is not addressed, which undermines the credibility of the analysis. The deforestation should be an important part of this paper, but the authors have even no introductions to it.

[Response] The deforestation dataset is an essential part of our study, and we have included a description of it in the Introduction. Regarding the time periods, when establishing the regression between changes in forest cover and precipitation (Fig. 4 and Fig. 5), we used their common observational period (i.e., 1982-2016) to avoid the effects of temporal mismatch. As a sensitivity analysis, we performed an additional investigation and found that the difference in precipitation change between the two periods (1980-2019 versus 1982-2016) is minor (New Supplementary Fig. 3, now as Fig. R3, versus Fig. 1 in the main paper). To address these two issues, we now write: “*Our key forest cover dataset spans from the years 1982 to 2016*⁵. To

align with the years of this coverage, we made these years a common observational period of forest cover and precipitation in the subsequent analysis, noting that precipitation trends show only minor differences during the two periods used (1980-2019 versus 1982-2016; Fig. 1 versus Supplementary Fig. 3)” (Lines 181-185).

Furthermore, we have also added more details on the forest cover dataset in the Methods section: “Forest cover was a fundamental dataset used in our analysis. Long-term forest cover was adopted from the Global Land Change dataset⁵. The dataset was produced by combining optical observations from multiple satellite sensors with a resolution of $0.05^{\circ} \times 0.05^{\circ}$ and for data covering a period of 35 years (1982-2016). Trees are defined as all vegetation taller than five meters in height. For our analysis, forest cover is defined as the fraction of a grid covered by the vertical projection of tree crowns^{5,6}” (Lines 547-552).

Fig. R3. (plot appears as Supplementary Fig. 3 in our new version) **Observed precipitation trend and its moisture sources for the Amazon.** The same as Fig. 1 but calculated for the period 1982-2016. **a**, Precipitation trend in the GPCP dataset. **b**, Precipitation trend in the GPCC dataset. The horizontal black line, at the latitude of 7.5°S , indicates our division between the northern and southern Amazon basins, while the outer black curve defines the full spatial extent of the Amazon basin. Stippling is for locations where the trend is statistically significant ($p < 0.05$). **c**,

Precipitation trend averaged over the whole, northern and southern Amazon basins for the two precipitation datasets. Error bars represent the standard errors of the trends. Asterisks indicate that the trend is significant ($p < 0.05$). **d**, Direct oceanic contributions to precipitation trend (P_{oceanic}). **e**, Terrestrial recycled contributions to precipitation trend (P_{recycled}). **f**, Oceanic and terrestrial recycled contributions to precipitation trends averaged over the whole, northern and southern Amazon basins. In all panels, all trends are calculated for the period 1980-2019 inclusively. In **d-f**, P_{oceanic} and P_{recycled} are derived from atmospheric moisture tracking based on the GPCP dataset. Here $P_{\text{total}} = P_{\text{recycled}} + P_{\text{oceanic}}$.

[Reviewer #2 Major Comment 7]

7. Weak background and insufficient framing of key concepts. The last paragraph of the introduction fails to clearly state the paper's objectives. The authors need to better explain: Concepts like "recycled rainfall," "upwind moisture," and "weighted forest cover" The novelty and importance of applying water vapor tracking models in studying deforestation–rainfall feedback. Relevant literature on vegetation feedbacks and atmospheric moisture recycling Terms such as "terrestrial recycled precipitation," "local and regional precipitation," "upwind deforestation," and "evapotranspiration" deserve dedicated background discussion. In contrast, the "Amazon dieback" discussion should be limited to one paragraph each in the introduction and the implications.

[Response] Thank you for these very helpful suggestions. We now clarify at the beginning of the paper where there was a gap in understanding, and from this, we provide pointers to the subsequent analysis (e.g., see response to Major Comment #2). We have now better supported the background of vegetation-rainfall feedbacks, included the relevant literature, and emphasised the novelty and importance of atmospheric moisture tracking in our study. Please see our response to your Major Comment #2, where we present our substantially rewritten Introduction and Conclusions. References to the potential for eventual Amazon dieback are now limited to a very brief mention in the Introduction and Discussion sections.

We have worked to use more frequently the definitions most commonly used in the literature. For instance, for precipitation recycling issues, we added the explanation as: "*To quantify the effects of Amazon moisture recycling, algorithms must account for the complex spatial*

connections between forest-derived moisture sources and precipitation sinks across the region” (Lines 41-43).

For terrestrial recycled precipitation, we have added: *“terrestrial recycled precipitation (i.e. land-sourced precipitation)” (Lines 107-108).*

For weight forest cover, we presented more information on its calculation: *“The FC_w variable integrates the satellite-derived forest cover within the combined local and upwind land region of moisture sources. This integration is weighted by the proportion of land moisture contribution at each upwind location to terrestrial recycled precipitation that falls at each local grid point (Methods). Hence, changes in FC_w value capture the full impact of forest cover change, including both locally and upwind, on local precipitation changes” (Lines 174-179).*

For upwind deforestation, we have added: *“upwind deforestation (i.e. deforestation in upwind regions that influences local rainfall levels)” (Lines 171-172).* Other technical terms are placed by common vocabulary.

In the last paragraph, we have clearly stated the objectives of our study: *“The objectives of our study are to explain features of precipitation changes over the Amazon basin and to investigate whether some of the observed changes are linked to direct forest cover change, i.e. deforestation” (Lines 53-55).*

[Reviewer #2 Major Comment 8]

8. Insufficient methodological detail L566: The description of the WAM-2layer model is overly brief. Key missing details include: What region is “tagged”? All land pixels or only the Amazon Basin? If only Amazon is tagged, how are “recycled” vs. “oceanic” precipitation defined? If all terrestrial pixels are tagged, how are Amazon effects isolated? The distinction between “P_recycled” and “P_oceanic”: Is $P_{total} = P_{recycled} + P_{oceanic}$? How is atmospheric moisture in transit handled? Is the model 2D or 3D in tracking water vapor? These fundamental methodological settings must be clarified.

[Response] Thank you for requesting additional information. To clarify how we set up the transport model, including definitions and details related to its operation, we have significantly expanded this section, carefully writing technical details as:

“Atmospheric moisture source and sink tracking

We used a well-established atmospheric moisture tracking model, the Water Accounting Model-2layers (WAM-2layers)²¹, to disentangle the oceanic versus terrestrial moisture contributions to the observed precipitation trend across the Amazon basin (see Supplementary Table 2 for an overview of the model). Here ocean-sourced precipitation (P_{oceanic}) is defined as the land precipitation that is contributed by moisture from the ocean evaporation, while land-sourced recycled precipitation (P_{recycled}) is defined as land precipitation that is contributed by moisture from terrestrial evapotranspiration. So observed precipitation (P_{total}) is the sum of land-sourced and ocean-sourced precipitation (i.e. $P_{\text{total}} = P_{\text{oceanic}} + P_{\text{recycled}}$). The WAM-2layers is an 2D offline moisture tracking model based on an Eulerian framework, and it quantifies the moisture source-sink relation between precipitation and evapotranspiration by tracking atmospheric moisture forward or backward in time¹⁶. The major model input includes reanalysis data from the updated ECMWF ERA5 database at a spatial resolution of $1^\circ \times 1^\circ$ for the period 1979-2020. All ERA5-based input variables are 6-hour gridded data (vertical specific humidity, zonal and meridional wind speeds, and surface pressure) except for precipitation and evapotranspiration which have a 1-hour temporal resolution. In each 15-min timestep, WAM-2layers solves the water balance of “tagged” moisture in an upper and lower layer in each atmospheric column, and the dynamic and transport of moisture between grids. Because the precipitation inside the Amazon basin can also be affected by forest cover changes outside the basin through cross-regional atmospheric moisture, we tagged moisture from all terrestrial grids to account for moisture changes from both the Amazon basin and outside. This model has proven to perform well against an online fully-3D tracking method⁶³” (Lines 583-605).

To assist readers who want a more detailed understanding of the moisture-tracking model and the key processes it addresses, we have also included an overview table that lists elements such as its resolution. Then, in the lower half of the table, we outline the main advantages and

disadvantages of using the WAM-2layers formulation in moisture tracking. This table is placed in the Supplementary Information.

Table R1. (*Supplementary Table S2 in our new version*) **Characteristics and (dis)advantages of WAM-2layers.**

Characteristics	Details
Framework	Eulerian approach (grid-based, 2D model)
Vertical resolution	Two-layer atmospheric model
Key equations	Atmospheric moisture budget (balance between evaporation, precipitation and moisture flux convergence at each time step)
Input data	1-hour precipitation and evapotranspiration (here constrained by observations); 6-hour specific humidity, zonal and meridional wind speeds at different pressure levels (from 1000 hPa to 100 hPa), and surface pressure from ERA5
Application	Tracks precipitation moisture sources and evapotranspiration sink (terrestrial vs. oceanic)
Time step	15 min
Spatial resolution	$1^\circ \times 1^\circ$
(Dis)advantages	Details
Computational efficiency	More efficient than Lagrangian models, suitable for large-scale and long-term simulations
Vertical simplification	Two-layer structure reduces complexity while capturing key vertical dynamics; it assumes well-mixed moisture within layers, but may suffer from imperfect vertical mixing especially in local-scale and short-term atmospheric processes
Tracking precision	Less effective than Lagrangian models in tracing specific moisture parcels over long distances or

[Reviewer #2 Major Comment 9]

9. Figure 1 shows that while P_recycled is decreasing (linked to deforestation), P_oceanic is increasing. However, the paper focuses solely on the decreasing recycled component. This selective focus becomes problematic in Figure 4 and in the discussion, where only P_recycled is extrapolated for future projection, which misrepresents the overall hydrological trend and exaggerates the dieback implications.

[Response] We appreciate this comment, and we should have clarified things more clearly. Our analysis does concentrate on the reduction in rainfall caused solely by deforestation in the southern Amazon basin. However, during the historical period at least, the increase in ocean-sourced precipitation is mainly concentrated in the northern Amazon basin. For the southern basin, oceanic-driven changes are far smaller than land-sourced precipitation change (Fig. 1f). Hence, focusing solely on recycled precipitation accounts for most of the changes experienced in the southern basin during the historical period.

Regarding future projections, we certainly agree with the position of the reviewer. Future overall downwind effects may need to account for more significant increases in rainfall transported from the oceans due to a warmer world that causes greater oceanic evaporative losses.

Based heavily on this comment, and our two response paragraphs above, we have added to the Discussion: *“We note that our quantification of projected precipitation reduction is solely based on changes in terrestrial recycled precipitation and its connection to deforestation over the past decades. Although the changes in the contribution of ocean-sourced moisture to observed precipitation alterations are minor in the historical period and for the southern basin (Fig. 1d, f), there may be more considerable change in the future. In a world much warmer than the present, increasing ocean-sourced moisture could also reshape the patterns of overall precipitation^{50,51}, which may also affect the fate and timing of the remaining Amazon forest”*

(Lines 325-332).

[Reviewer #2 Minor Comment 1]

1. L486: Clarify why El Niño impacts are discussed for the year 2020.

[Response] The common observational period for the GPCP and GPCC datasets is from the years 1980 to 2020. However, the El Niño event of year 2020 was so extreme that we decided not to include it in this current analysis. However, we are eager to emphasise that point clearly to any reader, and so we now write: *“Because trends in precipitation averaged over the whole, northern and southern Amazon basin during 1980-2020 were substantially (under)overestimated (-17%~117%) compared to that during periods 1980-2019 or 1980-2018, indicating abnormal high impact of year 2020 (El Niño year) on the long-term precipitation trend (Supplementary Fig. 7), we limited our study period to 1980-2019”* (Lines 499-503).

[Reviewer #2 Minor Comment 2]

2. L614–615: The phrase “... LAI increases and non-significant LAI decreases.” is unclear — please revise.

[Response] We have revised this sentence as: *“LAI increases or non-significant LAI decreases”* (Line 657).

[Reviewer #2 Minor Comment 3]

3. L474–487: Why were only two rainfall datasets used? Are they sufficient to support robust conclusions?

[Response] We selected these two specific precipitation datasets because they are satellite- or station-based, and moreover, because they have been proven to perform especially well over the Amazon basin (Sun et al., 2018; Smith et al., 2023) or the tropics in general (Trenberth et al., 2013). Although this does not necessarily provide additional validation, we note that these two particular datasets have also been widely used in previous studies of vegetation-climate feedbacks and the impact of deforestation on precipitation over the Amazon basin (Zemp et al., 2017; Xu et al., 2022; Bochow & Boers, 2023). We found that the two precipitation datasets

exhibited high consistency with each other in precipitation trends across both the northern and southern Amazon basins and in the moisture source attribution (Fig. 1 and Supplementary Fig. 2). Site-based observations also support these trends (Haghtalab et al., 2020).

We bring these points together by adding in the manuscript: *“In general, the two precipitation datasets compare well with each other (Fig. 1a versus Fig. 1b), while the remaining local differences in trends are likely related to their differing spatial resolutions and data sources^{12,13,23}. Furthermore, both the magnitude and the contrasting north-south pattern of these two precipitation trends generally agree well with other analyses using gauge-based observations²⁴”* (Lines 85-89).

[Reviewer #2 Minor Comment 4]

4. Figure 4: The use of linear extrapolation should be clearly stated and caveated.

[Response] Following your suggestion, we now clearly state and caveat the uncertainty of substantial linear extrapolation beyond the range of the regression that has been calibrated against historical data. We write: *“Although we establish a linear regression between precipitation reduction and forest cover loss at a spatial scale based on data from the past 35 years (Fig. 4), the relationship could become nonlinear at much higher amounts of forest cover loss^{12,23,27,34,49}. At higher deforestation scenarios, the decline in precipitation may be amplified due to either a stronger local self-reinforcing feedback mechanism that accelerates the suppression of recycled rainfall, or rainfall thresholds are crossed, triggering a nonlinear physiological response such that the remaining forest approaches dieback more rapidly. In this context, our projected precipitation reduction at the end of the 21st century may be underestimated”* (Lines 317-325).

[Reviewer #2 Minor Comment 5]

5. Figure 1: “mm yr⁻²” — clarify this unit. It appears inconsistent with “mm yr⁻¹ per year” in the main text. Use consistent and standard units throughout.

[Response] We have changed units to common “mm yr⁻¹ per year” throughout the text and

figures.

[Reviewer #2 Minor Comment 6-9]

6. Figure S4: Label the panels (A, B, C).
7. Figure S6: Label panels A and B.
8. Figure S7: Use consistent labels: panel (a), (b).
9. Figure S12: Label panels A, B, C, D.

[Response] Thank you for pointing out these typos. We have corrected all of them as you suggested. We also want to reiterate that during the process of generating a new manuscript version, we re-scanned the entire document to check for any additional minor grammatical errors that might have been present.

References

- Bochow, N.& Boers N. (2023), The South American monsoon approaches a critical transition in response to deforestation, *Sci Adv*, 9(40), eadd9973, doi:10.1126/sciadv.add9973.
- Cui, J. P., Lian X., Huntingford C., Gimeno L., Wang T., Ding J. Z., He M. Z., Xu H., Chen A. P., Gentine P., Piao S. L. (2022), Global water availability boosted by vegetation-driven changes in atmospheric moisture transport, *Nature Geoscience*, 15(12), 982–988, doi:10.1038/s41561-022-01061-7.
- Haghtalab, N., Moore N., Heerspink B. P., Hyndman D. W. (2020), Evaluating spatial patterns in precipitation trends across the Amazon basin driven by land cover and global scale forcings, *Theoretical and Applied Climatology*, 140(1-2), 411-427, doi:10.1007/s00704-019-03085-3.
- Smith, C.& Baker J. C. A.& Spracklen D. V. (2023), Tropical deforestation causes large reductions in observed precipitation, *Nature*, doi:10.1038/s41586-022-05690-1.
- Spracklen, D. V.& Arnold S. R.& Taylor C. M. (2012), Observations of increased tropical rainfall preceded by air passage over forests, *Nature*, 489(7415), 282-285, doi:10.1038/nature11390.
- Sun, Q., Miao C., Duan Q., Ashouri H., Sorooshian S., Hsu K.-L. (2018), A Review of Global Precipitation Data Sets: Data Sources, Estimation, and Intercomparisons, *Reviews of Geophysics*, 56(1), 79-107, doi:10.1002/2017rg000574.

- Trenberth, K. E., Dai A., van der Schrier G., Jones P. D., Barichivich J., Briffa K. R., Sheffield J. (2013), Global warming and changes in drought, *Nature Climate Change*, 4, 17, doi:10.1038/nclimate2067.
- Xu, X., Zhang X., Riley W. J., Xue Y., Nobre C. A., Lovejoy T. E., Jia G. (2022), Deforestation triggering irreversible transition in Amazon hydrological cycle, *Environmental Research Letters*, 17(3), 034037, doi:10.1088/1748-9326/ac4c1d.
- Zemp, D. C., Schleussner C. F., Barbosa H. M., Hirota M., Montade V., Sampaio G., Staal A., Wang-Erlandsson L., Rammig A. (2017), Self-amplified Amazon forest loss due to vegetation-atmosphere feedbacks, *Nat Commun*, 8, 14681, doi:10.1038/ncomms14681.

Point-to-point responses to reviewer comments

We thank the two reviewers for their particularly thorough reviews and helpful comments and suggestions, all of which have contributed to the improvement of the manuscript. Below are the comments from the reviewers, followed by our detailed, indented responses in blue. Additional or altered text in the revised manuscript, when cited in this response document, is marked in *dark grey italics*. The line numbers referred to relate to the “clean” non-tracked version of the revised manuscript.

Reviewers' comments:

Reviewer #1 (Remarks to the Author):

The authors have addressed all of my concerns. The manuscript is now ready for publication.

[Response] Thank you again for your original valuable suggestions that have all helped towards enhancing our manuscript. We are pleased to hear that our additional analyses have addressed all your concerns successfully.

Reviewer #1 (Remarks on code availability):

The *.mat files should be included in the public repository so that the results are reproducible.

[Response] Following your suggestions, we have uploaded all the underpinning *.mat files to a public repository: <https://doi.org/10.6084/m9.figshare.29649002.v2>. They are therefore fully accessible to any researcher.

Reviewer #2 (Remarks to the Author):

The authors have made some textual revisions and provided language-related responses, however these changes lack substantive improvements on several key points. After carefully reviewing the revised manuscript and the authors' responses to each comment, I must flag three critical issues that, in my view, fundamentally undermine both the reliability of the study and the validity of its conclusions:

1. Insufficient and speculative linkage between current results and historical deforestation. The manuscript attempts to link the present results to past deforestation events. However, this argument lacks robust quantitative evidence and relies largely on speculative reasoning. The method used to derive recycled precipitation is disconnected from deforestation, yet the second part of the manuscript makes a subjective determination that the results are induced by deforestation based on “visual correlations”. This way does not provide a sound methodological basis for establishing causality. I do not agree with this subjective judgment. As this point is central to the article, I

believe that, from this perspective alone, I can no longer support the manuscript.

[Response] We have taken this comment very seriously, and it has led us to conduct a final series of simulations that are process-based (i.e., derived from solving underlying equations). This provides a much stronger confirmation of a causal link between deforestation and reduced land-recycled moisture, which in turn lowers rainfall levels downstream. This additional information is shown in a new panel, Figure 4b (panel b of Fig. R2 below).

We also recognise the need to clarify how we develop our case for causality in the manuscript, and these modifications are outlined below. We acknowledge there were ambiguities in the original wording. We present our response to the request through three closely-related arguments, with the first answering the key issue around causality.

(1) Directly connecting upwind deforestation to recycled precipitation changes

Research confirms the link between land-surface changes and precipitation through moisture recycling (i.e. evapotranspiration) using moisture-tracking techniques (e.g. Spracklen et al., 2012; Staal, 2018; Hoek van Dijke et al., 2022; Cui et al., 2022; Bochow & Boers, 2023). We advance this method substantially by introducing an aggregated metric of weighted forest cover (FC_w) that accounts for both local and upwind forest loss simultaneously. In our initial paper version, we used a process-based atmospheric tracking model to derive, for each location, the impact of changes in combined local and upwind terrestrial recycling on rainfall. This second calculation was independent of prescribed land cover changes, allowing for an independent comparison, via correlation (original Fig. 4). We find statistical significance, but the reviewer correctly comments that this is not as convincing as process-led verification.

To validate the causality in Fig. 4 (generating new panel b; shown as Fig. R2 below), we undertook additional process-based experiments. We used forest cover changes during the past 35 years (1982-2016) and estimated deforestation-induced evapotranspiration (ET) changes following a standard forest cover-ET scaling method (Hoek van Dijke et al., 2022; Zan et al., 2024). We then conducted two sets of experiments using our moisture-tracking model with different ET as inputs: one with deforestation-induced ET changes and one without them. The difference in the moisture-tracking results demonstrates the impact of deforestation on recycled precipitation through ET. This process-driven calculation supports our initial statistically-based Fig. 4, directly addressing the reviewer's concern.

The cover changes, estimates of ET changes and the tracking-derived precipitation changes are shown below in Fig. R1a-c respectively. We can then compare the aggregated (so local and

upwind) effects of deforestation in our aggregated bulk parameter, FC_w, against these recycled rainfall estimates (Fig. R1c). We present these model-derived calculations, across land points, as a new “x-y” Fig 4b (Fig. R2b below).

The similarities between Fig. R2a, which is our original diagram, and Fig. R2b, provide process-led confirmation. In summary, the new Fig. R2b instead calculates rainfall changes based on atmospheric calculations forced by direct estimates of surface ET change. This supports our initial calculations, which use land recycled estimates in which ET changes are implicit, relying on a final correlation analysis to confirm that rainfall decreases link to deforestation (Fig. 2a). It is of interest that the new calculations of Fig. 2b estimate a lower sensitivity (smaller gradient) compared to our data-led Fig 2a, which may be due to more local land-atmosphere feedbacks, or an underestimate of evaporative changes. Both of these issues present interesting challenges for future research and could influence measurement campaigns. We again thank the reviewer for encouraging this additional analysis, as it might provide further reasons for our paper to assist others.

Related to the revised Fig. 4, in the main text, we have added this validation of the causality:

“To validate the causality of the link between recycled precipitation and FC_w presented in Fig. 4a, we conduct additional process-based experiments using our moisture-tracking model. We first estimate deforestation-induced evapotranspiration changes based on a forest cover-evapotranspiration scaling approach²⁰. We then use these estimates, alongside estimates of evapotranspiration without deforestation, to directly drive the moisture-tracking model. The difference between these two simulations represents the causal impact of deforestation on recycled precipitation, now calculated directly by evapotranspiration, which represents moisture recycling. The results also show a strong declining (i.e., negative) relationship between recycled precipitation and FC_w, supporting our more observational-based results (Fig. 4a). The weaker impact of deforestation on recycled precipitation in Fig. 4b, illustrated by the lower gradient of the fitted regression line of Fig 4b, may be related to the inclusion of only the direct impact of deforestation on evapotranspiration, thus ignoring local feedbacks where atmospheric processes further suppress evapotranspiration and hence precipitation. The differences may also stem from an underestimate of deforestation impacts on evapotranspiration. Understanding these differences in sensitivity may guide future insights or measurement campaigns to better constrain local evapotranspiration changes following

deforestation” (Lines 223-239).

Fig. R1. Changes in forest cover and its impact on evapotranspiration (ET) and recycled precipitation (P_recycled). **a**, Changes in forest cover during the period of years 1982-2016. **b**, Calculated changes in ET related to the forest cover changes of panel **a**. **c**, Forest cover-induced changes in P_recycled, derived from an atmospheric transport model forced with evapotranspiration estimated with and without deforestation, and with the difference in these drivers in panel **b**.

Fig. R2. (plot appears as Fig. 4 in our new version, the major change being the addition of new panel b) **The impacts of forest cover change on recycled precipitation.** **a**, Correlation between weighted forest cover and terrestrial recycled precipitation in the southern Amazon basin. Recycled precipitation is derived from the moisture-tracking model driven by water balance-based evapotranspiration. Regression line (black line) is based on different spatial points, with each point representing local changes in recycled precipitation and weighted forest cover in the southern Amazon basin for the common period 1982-2016. Each point represents a $1^\circ \times 1^\circ$ gridbox within the southern basin. The blue, red and green lines mark the changes in weighted forest cover in the past 35 years, SSP2-4.5 (primf) and SSP2-4.5 (primf + secdf) scenarios, respectively, and the corresponding reductions in terrestrial recycled precipitation. “Primf” represents primary forested land, while “secdf” represents secondary forested land including forest regrowth and climate mitigation strategies such as afforestation and reforestation (Methods). The shaded areas denote the 95% confidence intervals of changes in the southern basin. For illustration purposes, the horizontal and vertical zero lines are shown as grey dashed lines. **b**, The same as **a**, but instead, the level of changes in precipitation caused by altered land moisture recycling is derived from the difference between projections of the moisture-tracking model when driven directly by evapotranspiration estimates with and without

deforestation. The evapotranspiration post-deforestation was based on a forest cover-evapotranspiration scaling approach. As panel **b** is direct process model output, we do not present this as a statistical finding (e.g. with p value), but we do fit a linear regression line (black line) to aid comparison with panel **a**.

(2) Development of a new observation-based ET product

Robust estimates of ET are key to capturing the signals of deforestation and to accurately force our atmospheric moisture transport model. We used multiple sets of observational data, including satellite-derived precipitation, gauge-based discharge, and satellite-derived terrestrial water storage to estimate ET based on the water balance approach (Swann & Koven, 2017; Baker et al., 2021). This newer calculation has been compared to other ET products (Fig. R3 below) and at individual sites (Supplementary Fig. 10). The comparison indicates that some other ET products underestimate recent ET reductions in the southern Amazon basin (Fig. R3).

In the objectives, we now write: “Unlike most previous studies, we use satellite-based estimates of precipitation and, furthermore, develop a new water balance-based estimate of historical evapotranspiration as required as a forcing to the atmospheric moisture tracking framework. Creating the latter dataset involves refining satellite-based estimates of evapotranspiration and is therefore strongly guided by multiple measurement sets. Both datasets more accurately constrain this driver to our tracking framework (Methods), reducing uncertainties in trends that may exist in other reanalysis precipitation and evapotranspiration datasets²². More importantly, for the analysis presented here, the water balance-based evapotranspiration more precisely captures the signals of deforestation (Methods). This enhancement better supports projections of local and upstream deforestation-induced changes on moisture transport and precipitation” (Lines 78-88).

In the Methods section, we illustrate the advantages for process representation by using our water balance-based ET estimates compared to existing evapotranspiration products: “The ET_{wb} values showed a more pronounced time-evolving reduction than the latter two datasets (Supplementary Fig. 9). Such larger decreasing trends are consistent with the observed precipitation changes (Fig. 1a, b) and supported by previous observation-based estimates of evapotranspiration^{47,58}. At the site level, we also confirmed that our ET_{wb} estimates outperformed the GLEAM and FLUXCOM data products for most locations where point data is available (Supplementary Fig. 10). These advancements in ET_{wb} provided a more solid basis for tracking the dynamics of atmospheric moisture transport, by offering more reliable surface boundary conditions to such models.” (Lines 581-588).

Fig. R3. (plot appears as Supplementary Fig. 9 in our new paper version) **Comparison of water balance-based evapotranspiration (ET_wb) with GLEAM and FLUXCOM datasets over the Amazon basin.** **a-b**, Climatological annual mean (**a**) and trend (**b**) in ET, averaged spatially over the Amazon basin. Error bars represent the standard errors of the means or trends. **c-h**, Spatial patterns of climatological annual mean (**c**, **e** and **g**) and trend (**d**, **f** and **h**) in ET for ET_wb, GLEAM and FLUXCOM estimates as marked. Stippling are locations where the trends are statistically significant ($p < 0.05$).

(3) Check that our key result is not “by chance” and that autocorrelation is not impacting findings

To add further robustness to our findings, we note that we have performed checks of our regression analysis between recycled precipitation and the deforestation metric FC_w, as shown in Fig. 4a. In one particular test, we randomly shuffled the locations of the grids of known forest cover and its changes. We then used this randomised data to recalculate a time-evolving FC_w statistic at each location, which we subsequently compared against local changes in recycled rainfall. This process is illustrated below, via panels a-to-d of Fig. R4, with Fig. R4d having an equivalent format to Fig. 4a in the main manuscript. With deforestation data randomised, our regression is lost (Fig. R4d), indicating that our headline finding, that local and upwind deforestation reduces rainfall, could not have been derived by chance.

Overall, we hope that all the additional experiments and analyses above demonstrate robustness in our causal link between deforestation and Amazonian rainfall changes. We again apologise for the phrase “visual correlation,” which is not a substitute for the rigorous statistical testing we carried out. We have removed these words to prevent confusion.

Fig. R4. (plot appears as Supplementary Fig. 12 in our new paper version) **Evaluation of autocorrelation impact on the relationship between changes in weighted forest cover and recycled precipitation.** Panel a shows the mean of forest cover data, which also evolves over time, at each location. Panel b is the spatial randomisation of the data in panel a, while

preserving, for each moved point, the original timeseries. Panel c displays the trend in FC_w, derived from the randomised ('r') data, and so named rFC_w. Panel d is of identical format to Fig. 4a, except that the "x" variable is rFC_w. The blue arrows and labels indicate the data processing in each step between subplots.

2. High methodological risk in projecting a weak historical linkage linearly into the future. The manuscript extrapolates a very weak historical relationship between deforestation and precipitation directly into future climate scenarios using a linear approach. This practice poses significant scientific risks and undermines the credibility of the projections.

[Response] We fully understand this comment. First of all, we want to again recognise how your encouragement, in Comment#1 above, has strengthened the causal verification of the relationship between deforestation and precipitation. Although our regression has robustness (statistically significant; $p < 0.001$), this extra analysis strengthens our argument that local and upwind deforestation is lowering Amazonian rainfall. Therefore, with this evidence base, we believe that it is acceptable to extrapolate to relatively small additional changes in the climate (such as those future changes defined by SSP2.45). Small extrapolations are also reasonable because, as we have confirmed based on multiple climate model projections, the wind field changes are expected to be minor in SSP2.45 (Supplementary Fig. 5). This invariance ensures that the atmospheric tracking calibration remains valid. However, based on this request, we are pleased to have strengthened the caveats in the Discussion section.

We now write: "The projection of future precipitation change can only be considered as a first approximation. Although the relationship between recycled precipitation and FC_w is statistically significant (Fig 4a) (further supported by process modelling; Fig. 4b) it does still have sizeable noise, and therefore remaining uncertainty may be amplified with extrapolation" (Lines 369-373).

Based on this reviewer request (and similar requests at the initial reviewer assessment), we note the scientific risk of extrapolating much further into the future, as nonlinear effects may emerge that could deviate from any linear projection. We have amended this text and now write more clearly: *"Although we establish a linear regression, Fig 4a (with additional process modelling support; Fig 4b) between precipitation reduction and forest cover loss at a spatial scale based on data from the past 35 years (Fig. 4a), the relationship could become nonlinear at much higher amounts of forest cover loss^{12,23,27,34,49}. At much higher deforestation scenarios, the decline in precipitation may be amplified due to either a stronger local self-reinforcing feedback mechanism that accelerates the suppression of recycled rainfall. Additionally, key*

rainfall thresholds may be crossed, triggering a nonlinear physiological response such that the remaining forest approaches dieback more rapidly. In this context, projected precipitation reductions at the end of the 21st century may be underestimated, and especially scenarios involving very substantial continued Amazonian deforestation” (Lines 353-362).

3. Lack of clear distinction between total terrestrial moisture and Amazonian-sourced moisture. The manuscript does not clearly differentiate between “total terrestrial moisture” and “Amazonian-sourced moisture.” Given the stated research aims and study design, the analysis should explicitly tag and track Amazonian-sourced moisture, rather than using the broader “terrestrial moisture” metric. This misuse creates logical inconsistencies and may fundamentally alter the interpretation of the results. For example, by tagging all land grids, the calculated recycled precipitation is artificially elevated, because it includes contributions from land areas outside the Amazon basin. This additional portion has no connection to the Amazonian moisture, deforestation, and future deforestation, yet the authors interpret and present it as “Amazonian-sourced moisture” and use them for attribution explanation and future projection thereby conflating two distinct concepts.

[Response] *Since most deforestation has occurred in the Amazon basin, we inadvertently implied that these were the only Amazon points included in the analysis. We apologise for our imprecise terminology; atmospheric tracking includes all land points, so accounting for all terrestrial moisture sources. We have standardised and reworded as “terrestrial recycled precipitation” or “land-sourced precipitation” throughout the manuscript.*

If of interest, and aside from our notation error above, this reviewer’s comment also prompted us to examine the paper further for any risk of ambiguity. We also compare our analyses with existing research, in the context of the completeness of tracking water released from the land surface. The purpose of our study is not to isolate the impact of deforestation solely within the Amazon basin on precipitation. Instead, we aim to identify the role of all deforestation across much of South America, including areas outside the Amazon basin, on historical “observed” precipitation. This differentiates our analysis from previous coupled land-atmosphere model-based studies (Sampaio et al., 2007; Khanna & Medvigy, 2014; Lejeune et al., 2015; Ruiz-Vásquez et al., 2020; Qin et al., 2025), which are often prescribed with observed or assumed deforestation scenarios limited to the Amazon basin. Although land-sourced precipitation primarily originates from the Amazon basin, upwind land (outside the basin) contributes to approximately one-third of the total precipitation, validating the point made by the reviewer (Fig. R5). That said, although we do simulate changes across South America, the deforestation data will reflect where most land use has occurred, which is in the Amazon basin.

To capture these issues of land locations included, we have added to the paper: “*This*

comparison enables a more rigorous assessment of how both local and upwind forest cover loss, including any over a large geographical range outside the Amazon basin, impacts local precipitation. Our approach enables the creation of a new metric, weighted forest cover, which quantifies these effects (Methods). The aim of introducing this metric is to capture all the impacts of any deforestation within and outside the Amazon basin, rather than isolating the impact on rainfall from deforestation within the Amazon basin alone. In general, though, most deforestation to date has been within the Amazon basin” (Lines 72-78).

Fig. R5. Moisture sources of precipitation in the Amazon basin. a, Climatological mean (years 1980-2019 inclusive) precipitation moisture sources, from either the land or ocean, dependent on location. Arrows show the mean surface wind fields. The regions marked with coloured text correspond to the histogram bars in panel b. b, Precipitation averaged over the different regions shown in a. Hence, in both panels, land-sourced precipitation is partitioned into contributions from the Amazon and upwind land regions.

References

- Baker, J. C. A., Garcia-Carreras L., Gloor M., Marsham J. H., Buermann W., da Rocha H. R., Nobre A. D., de Araujo A. C., Spracklen D. V. (2021), Evapotranspiration in the Amazon: spatial patterns, seasonality, and recent trends in observations, reanalysis, and climate models, *Hydrology and Earth System Sciences*, 25(4), 2279-2300, doi:10.5194/hess-25-2279-2021.
- Cui, J. P., Lian X., Huntingford C., Gimeno L., Wang T., Ding J. Z., He M. Z., Xu H., Chen A. P., Gentine P., Piao S. L. (2022), Global water availability boosted by vegetation-driven changes in atmospheric moisture transport, *Nature Geoscience*, 15(12), 982-988, doi:10.1038/s41561-022-01061-7.
- Hoek van Dijke, A. J., Herold M., Mallick K., Benedict I., Machwitz M., Schlerf M., Pranindita A., Theeuwens J. J. E., Bastin J.-F., Teuling A. J. (2022), Shifts in regional water availability due to global tree restoration, *Nature Geoscience*, 15(5), 363-368, doi:10.1038/s41561-022-00935-0.
- Khanna, J. & Medvigy D. (2014), Strong control of surface roughness variations on the simulated dry season

- regional atmospheric response to contemporary deforestation in Rondônia, Brazil, *Journal of Geophysical Research: Atmospheres*, 119(23), 13,067-013,078, doi:10.1002/2014jd022278.
- Lejeune, Q., Davin E. L., Guillod B. P., Seneviratne S. I. (2015), Influence of Amazonian deforestation on the future evolution of regional surface fluxes, circulation, surface temperature and precipitation, *Climate Dynamics*, 44(9-10), 2769-2786, doi:10.1007/s00382-014-2203-8.
- Qin, Y., Wang D., Ziegler A. D., Fu B., Zeng Z. (2025), Impact of Amazonian deforestation on precipitation reverses between seasons, *Nature*, 639(8053), 102-108, doi:10.1038/s41586-024-08570-y.
- Ruiz-Vásquez, M., Arias P. A., Martínez J. A., Espinoza J. C. (2020), Effects of Amazon basin deforestation on regional atmospheric circulation and water vapor transport towards tropical South America, *Climate Dynamics*, 54(9-10), 4169-4189, doi:10.1007/s00382-020-05223-4.
- Sampaio, G., Nobre C., Costa M. H., Satyamurty P., Soares-Filho B. S., Cardoso M. (2007), Regional climate change over eastern Amazonia caused by pasture and soybean cropland expansion, *Geophysical Research Letters*, 34(17), doi:10.1029/2007gl030612.
- Spracklen, D. V.& Arnold S. R.& Taylor C. M. (2012), Observations of increased tropical rainfall preceded by air passage over forests, *Nature*, 489(7415), 282-285, doi:10.1038/nature11390.
- Staal, A., Tuinenburg O. A., Bosmans J. H. C., Holmgren M., van Nes E. H., Scheffer M., Zemp D. C., Dekker S. C. (2018), Forest-rainfall cascades buffer against drought across the Amazon, *Nature Climate Change*, 8(6), 539-543, doi:10.1038/s41558-018-0177-y.
- Swann, A. L. S.& Koven C. D. (2017), A Direct Estimate of the Seasonal Cycle of Evapotranspiration over the Amazon Basin, *Journal of Hydrometeorology*, 18(8), 2173-2185, doi:10.1175/jhm-d-17-0004.1.
- Zan, B., Ge J., Mu M., Sun Q., Luo X., Wei J. (2024), Spatiotemporal inequality in land water availability amplified by global tree restoration, *Nature Water*, doi:10.1038/s44221-024-00296-5.

Point-to-point responses to reviewer comments

We thank Reviewer #2 for his/her additional suggestions, all of which have contributed to the improvement of the manuscript. Below are the comments from the reviewers, followed by our detailed, indented responses in blue. Additional or altered text in the revised manuscript, when cited in this response document, is marked in *dark grey italics*. The line numbers referred to relate to the “clean” non-tracked version of the revised manuscript.

Reviewer #2 (Remarks to the Author):

Though the authors make some serious mistakes, I believe this paper is still worthy of publication. However, they need to carefully address the two points raised below. I would also like to see the improvement once the paper is online:

1. In the response to my comment #3, the authors wrote: “The purpose of our study is not to isolate the impact of deforestation solely within the Amazon basin on precipitation. Instead, we aim to identify the role of all deforestation across much of South America, including areas outside the Amazon basin, on historical ‘observed’ precipitation.” If the purpose and results of this study is not specifically focused on the Amazon but rather on South America as a whole, then the title, abstract, and all relevant sections in the manuscript where “Amazon” is used should be revised accordingly.

[Response] Thank you for your comments. We are pleased that our work is described as “worthy of publication”, and we appreciate your further time spent reviewing the manuscript.

We understand the concern raised regarding the geographical scope of our analysis. We tracked changes in moisture linked to deforestation across all of South America. However, the reason the Amazon falls into focus across the diagrams is because that is where most land use change has occurred, particularly in the southern Amazon basin. Although many studies have documented widespread deforestation in the Amazon basin over recent decades, its feedback to precipitation has not been fully quantified at regional scale, and certainly not with simulation methods fully constrained and validated by multiple data streams, as we have undertaken.

To clearly explain why we focus on the Amazon basin, we have added: “*The aim of introducing this metric is to capture all the impacts of any deforestation within and outside the Amazon basin across the South America, rather than isolating the impact on rainfall from deforestation within the Amazon basin alone. We mainly focus on the Amazon basin because most deforestation to date has been within the basin*” (Lines 75-79). In the abstract, we have clarified the focus on the Amazon and its relation to South America: “*We discover that this reduction in precipitation is primarily (52-72%) related to widespread deforestation in the southern basin and upwind regions over South America*” (Lines 19-21). In addition, in other

places where available, we have added “*South American deforestation, the majority of which has occurred so far in the Amazon Basin*” or “*over South America*” to clarify that the impact of deforestation comes from all of South America.

2. My key concern is that the authors only addressed my comments superficially in the response letter by making language clarifications but made no substantive changes in the main text or results. This way leaves the manuscript with misleading statements and inconsistent conclusions, which undermines the scientific credibility of the work and makes the arguments increasingly self-contradictory. I understand that terms like “Amazon” and the deleted phrase “Amazon dieback” may sound more appealing to editors and readers, but scientific integrity should not be compromised for the sake of attractiveness.

[Response] We understand this comment. Before replying, we would like to clarify that we conducted additional simulations and made significant revisions to the main text and results, based on your requests, adding to the language clarifications. Your suggestions were really appreciated and helpful. Specifically, you guided us to: (1) perform a final series of process-based simulations to confirm the causal link between deforestation and precipitation, resulting in a new panel b for Figure 4 (which explains the connection between these two key data strands, initially compared statistically). This additional panel has led to new text describing the increased rigour in confirming causality. Your advice also helped us to: (2) demonstrate the benefits of improved process representation by using our water balance-based ET estimates, confirmed to outperform other evapotranspiration products, and (3) strengthen the caveats in the Discussion section, including more caution about the nonlinear effects of our results if they were ever extrapolated to higher levels of deforestation.

We apologise if this additional analysis was not sufficiently highlighted in our revised paper. To improve clarity, we have now explicitly marked the key textual and analytical revisions in this revised version (please see Lines 224-240, Lines 428-460 and Lines 341-377 in the main text).

Regarding the term “Amazon” versus “South America”, we have clarified, where appropriate, why we focused on the Amazon basin and have rephrased “*Amazon*” to “*South America*”. Please see our response to your comment #1 above in this context.